# Discovering group dynamics in coordinated time series via hierarchical recurrent switching-state models

**Michael T. Wojnowicz**[1,2,3]**, Kaitlin Gili**[1]**, Preetish Rath**[1]**,**
**Eric Miller**[1]**, Jeffrey Miller**[2]**, Clifford Hancock**[4]**, Meghan O'Donovan**[4]**,**
**Seth Elkin-Frankston**[1,4]**, Tad T. Brunye**[1,4]**, and Michael C. Hughes**[1]

[1] *Tufts University, Medford, MA, USA*

[2] *Harvard University, Boston, MA, USA*

[3] *Montana State University, Bozeman, MT, USA*

[4] *U.S. Army Combat Capabilities Development Command Soldier Center (CCDC SC), Natick, MA, USA*

**Reviewed on OpenReview:** *https://openreview.net/forum?id=LHchZthcOf*

## Abstract

We seek a computationally efficient model for a collection of time series arising from multiple interacting entities (a.k.a. "agents"). Recent models of temporal patterns across individuals fail to incorporate explicit system-level collective behavior that can influence the trajectories of individual entities. To address this gap in the literature, we present a new hierarchical switching-state model that can be trained in an unsupervised fashion to simultaneously learn both system-level and individual-level dynamics. We employ a latent system-level discrete state Markov chain that provides top-down influence on latent entity-level chains which in turn govern the emission of each observed time series. Recurrent feedback from the observations to the latent chains at both entity and system levels allows recent situational context to inform how dynamics unfold at all levels in bottom-up fashion. We hypothesize that including both top-down and bottom-up influences on group dynamics will improve interpretability of the learned dynamics and reduce error when forecasting. Our *hierarchical switching recurrent dynamical model* can be learned via closed-form variational coordinate ascent updates to all latent chains that scale linearly in the number of entities. This is asymptotically no more costly than fitting a separate model for each entity. Analysis of both synthetic data and real basketball team movements suggests our lean parametric model can achieve competitive forecasts compared to larger neural network models that require far more computational resources. Further experiments on soldier data as well as a synthetic task with 64 cooperating entities show how our approach can yield interpretable insights about team dynamics over time.

## 1 Introduction

We consider the problem of jointly modeling a collection of multivariate time series arising from individual entities that can influence each other's behavior over time and might share goals. Each series in the collection describes the evolution of one *entity*, sometimes also called an "agent" (Yuan et al., 2021). All entities are observed over the same time period within a shared environment or *system*. Our work is motivated by the need to capture an essential property of such data in many applications: the temporal behaviors of the individual entities are *coordinated* in a systematic but fundamentally latent (i.e., unobserved) manner.

As a motivating example, consider the dynamics of a team sport like basketball (Terner & Franks, 2021). To accomplish a team goal, one player might set a "screen," physically blocking a defender to allow a teammate an open drive to the basket. This screen could be preplanned or arise as players act on an in-moment opportunity. As another example, consider marching band players that practice moving together in a coordinated fashion

---

Open-source code: **https://github.com/tufts-ml/team-dynamics-time-series/**

across a field. Most movements are planned out by a coach. However, in some cases, situational adjustments are needed: when enough individuals make mistakes in their own trajectories, the coach may decide to rein in all of the players and reset. Our experiments directly cover basketball and marching band case studies later in Sec. 5.2 and Sec. 5.3, respectively. Similar group dynamics arise in many other domains, such as businesses in a common economic system (van Dijk et al., 2002), animals in a shared habitat (Sun et al., 2021), biological cells sharing copy-number mutations (Babadi et al., 2023), or students in a collaborative problem-solving session (Odden & Russ, 2019; Earle-Randell et al., 2023).

In all these examples, the dynamics of the individuals are far from independent. Instead, the observed trajectories exhibit "top-down" patterns of coordination that are planned in advance or learned from extensive training together. They also exhibit "bottom-up" adaptations of the individuals and the group to evolving situational demands. We seek to build a model that can infer how group dynamics evolve over time given only entity-level sensory measurements, while taking into account top-down and bottom-up influences.

While modeling individual time series has seen many recent advances (Linderman et al., 2017; Gu et al., 2022; Farnoosh et al., 2021), there remains a need for improved models for coordinated collections of time series. To try to model a collection of time series, a simple approach could repurpose models for individual time series. As a step beyond this, some efforts pursue *personalized* models that allow custom parameters that govern each entity's dynamics while sharing information between entities via common priors on these parameters (Severson et al., 2020; Linderman et al., 2019; Alaa & van der Schaar, 2019), often using mixed effects (Altman, 2007; Liu et al., 2011). But personalized models allow each sequence to unfold asynchronously without interaction. In contrast, our goal is to specifically model coordinated behavior within the same time period. Others have pursued this goal with complex neural architectures that can jointly model "multi-agent" trajectories (Zhan et al., 2019; Alcorn & Nguyen, 2021; Xu et al., 2022). Instead, we focus on parametric methods that are easier to interpret and more likely to provide sample-efficient quality fits in applications with only a few minutes of available data (such as the data described in Sec. 5.4).

One potential barrier to modeling coordination across entities is computational complexity. For instance, a model with discrete hidden states which allows interactions among entities has a factorial structure with inference that scales *exponentially* in the number of entities (see Sec. 3). In this paper we present a tractable framework for modeling collections of time series that overcomes this barrier. All estimation can be done with cost linear in the number of entities, making our model's asymptotic runtime complexity no more costly than fitting separate models to each entity.

The **first key modeling contribution** is an explicit representation of the *hierarchical* structure of group dynamics, modeled via a latent system-level state that exerts "top-down" influence on each individual time series. We use well-known switching-state models (Rabiner, 1989) as a building block for both system-level and individual-level dynamics. As shown in Fig. 1, our model posits two levels of latent discrete state chains: a system-level chain shared by all entities and an entity-level chain unique to each entity. We assume that the system-level chain is the sole mediator of cross-entity coordination; each entity-level chain is conditionally independent of other entities given the system-level chain. Our model achieves "top-down" coordination via the system-level chain's influence on entity-level state transition dynamics. In turn, an emission model produces each entity's observed time series given the entity-level chain.

The **second key modeling contribution** is to allow "bottom-up" adjustments to recent situational demands *at all levels* of the hierarchy. In our assumed generative model in Fig. 1, the next system-level state and entity-level state both depend on feedback from per-entity observed data at the previous timestep. In the basketball context, this feedback captures how a basketball player driving to the basket switches to another behavior after reaching their goal. Critically, with our model that situational change in one player can quickly influence the trajectories of other players. We refer to observation-to-latent feedback as *recurrent*, following Linderman et al. (2017). Note that this recurrent feedback also allows the model to learn state duration distributions that are more expressive than the geometric distributions that would be implied otherwise (Ansari et al., 2021). Previously, Linderman et al. (2017) incorporated such recurrent feedback into a model with a flat single-level of switching states. We show how recurrent feedback can inform a two-level hierarchy of system-level and entity-level states, so that entity-level observations can drive system-level transitions in a computationally efficient manner.

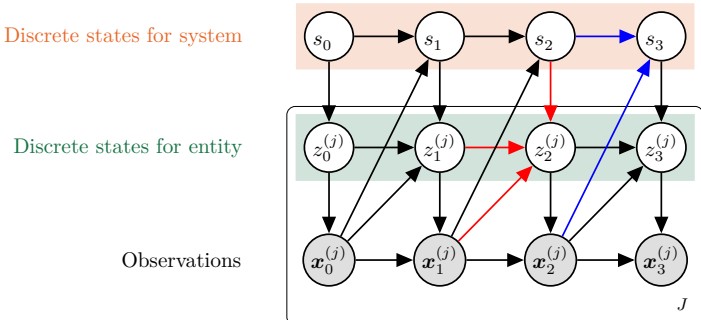

Figure 1: Graphical model representation of our *hierarchical switching recurrent dynamical model* (`HSRDM`) for a system of $J$ interacting entities. Colored edges highlight key insights behind our flexible transition models of system-level hidden states $s$ and entity-level hidden states $z$. Transitions to the next system state depend via blue arrows on the current system state and recurrent feedback from observations of all entities (up-diagonal). Transitions to the next entity state depend via red arrows on the next system-level state (down arrow), current entity state (horizontal), and recurrent feedback from that entity's observations (up-diagonal).

Our overall contribution is thus a proposed framework – *hierarchical switching recurrent dynamical models* – by which our two key modeling ideas provide a natural and *cost-effective* solution to the problem of unsupervised modeling of coordinated time series. Unlike other models, our framework allows each entity's next-step dynamics to be driven by both a system-level discrete state ("top-down" influence) and recurrent feedback from previous observations ("bottom-up" influence). Optional exogenous features (e.g. the ball position in basketball) can also be incorporated. We further provide a variational inference algorithm for simultaneously estimating model parameters and approximate posteriors over system-level and entity-level chains. Each chain's posterior maintains the model's temporal dependency structure while remaining affordable to fit via efficient dynamic programming that incorporates recurrent feedback. We conduct experiments on two synthetic datasets as well as two real-world tasks to demonstrate the model's superior capability in discovering hidden dynamics as well as its competitive forecasts obtained at low computational cost.

## 2 Model Family

Here we present a family of *hierarchical switching recurrent dynamical models* (`HSRDMs`) to describe a collection of time series gathered from $J$ entities that interact over a common time period (discretized into timesteps $t \in \{0, 1, 2, \ldots, T\}$) and in a common environment or *system*. For each entity, indexed by $j \in \{1, \ldots, J\}$, we observe a time series of feature vectors $\{\boldsymbol{x}_t^{(j)} \in \mathbb{R}^D, t = 0, 1, 2, \ldots, T\}$.

Our `HSRDM` represents the $j$-th entity via two random variables: the observed or "emitted" features $\boldsymbol{x}_{0:T}^{(j)}$ and a hidden entity-level discrete state sequence $z_{0:T}^j = \{z_t^{(j)} \in \{1, \ldots, K_j\}, t = 0, \ldots, T\}$. We further assume a system-level latent time series of discrete states $s_{0:T} = \{s_t \in \{1, \ldots, L\}, t = 0, \ldots, T\}$. The complete joint density of all random variables across all $J$ entities factorizes according to the graphical model in Figure 1 as

$$p(\boldsymbol{x}_{0:T}^{(1:J)}, z_{0:T}^{(1:J)}, s_{0:T} \mid \theta) = \underbrace{p(s_0 \mid \theta)}_{\text{system state init.}} \prod_{t=1}^{T} \underbrace{p(s_t \mid s_{t-1}, \boldsymbol{x}_{t-1}^{(1:J)}, \theta)}_{\text{system state transitions}} \cdot \prod_{j=1}^{J} \underbrace{p(\boldsymbol{x}_{0:T}^{(j)}, z_{0:T}^{(j)} \mid s_{0:T}, \theta)}_{\text{per-entity states and emissions}} . \quad (2.1)$$

Here, $\theta$ denotes all model parameters, and superscript $^{(1:J)}$ denotes the concatenation of variables over all entities. For a specific entity at index $j$, its individual states and emitted data factorize as

$$p(\boldsymbol{x}_{0:T}^{(j)}, z_{0:T}^{(j)} \mid s_{0:T}, \theta) = \underbrace{p(z_0^{(j)} \mid s_0, \theta)}_{\text{entity state init.}} \prod_{t=1}^{T} \underbrace{p(z_t^{(j)} \mid z_{t-1}^{(j)}, \boldsymbol{x}_{t-1}^{(j)}, s_t, \theta)}_{\text{entity state transitions}} \cdot \underbrace{p(\boldsymbol{x}_0^{(j)} \mid z_0^{(j)}, \theta)}_{\text{emission init.}} \prod_{t=1}^{T} \underbrace{p(\boldsymbol{x}_t^{(j)} \mid \boldsymbol{x}_{t-1}^{(j)}, z_t^{(j)}, \theta)}_{\text{emission dynamics}}.$$

The design principle of `HSRDMs` is to coordinate the switching-state dynamics of multiple entities so they receive top-down influence from system-level state as well as bottom-up influence via recurrent feedback from

entity observations (diagonal up arrows in Fig. 1). Under the generative model, the next entity-level state depends on the interaction of three sources of information: the next state of the system, the current state of the entity, and the current entity observation. Likewise, the next system state depends on the current system state and observations from *all* entities.

**Transition models.** To instantiate our two-level discrete state transition distributions, we use categorical generalized linear models to incorporate each source of information via additive utilities

$$s_t \mid s_{t-1}, \boldsymbol{x}_{t-1}^{(1:J)} \sim \text{Cat-GLM}_L\bigg( \underbrace{\widetilde{\boldsymbol{\Pi}}^T \boldsymbol{e}_{s_{t-1}}}_{\text{transition preferences}} + \underbrace{\boldsymbol{\Lambda}\, g_\psi\big(\boldsymbol{x}_{t-1}^{(1:J)}, \boldsymbol{v}_{t-1}\big)}_{\text{recurrent feedback}} \bigg), \tag{2.2a}$$

$$z_t^{(j)} \mid z_{t-1}^{(j)}, \boldsymbol{x}_{t-1}^{(j)}, s_t \sim \text{Cat-GLM}_K\bigg( \underbrace{(\widetilde{\boldsymbol{P}}_j^{(s_t)})^T \boldsymbol{e}_{z_{t-1}^{(j)}}}_{\text{transition preferences}} + \underbrace{\boldsymbol{\Psi}_j^{(s_t)} f_\phi\big(\boldsymbol{x}_{t-1}^{(j)}, \boldsymbol{u}_{t-1}^{(j)}\big)}_{\text{recurrent feedback}} \bigg). \tag{2.2b}$$

Across levels, common sources of information drive these utilities. First, the *state-to-state transition* term selects an appropriate log transition probability vector from matrices $\widetilde{\boldsymbol{\Pi}}, \widetilde{\boldsymbol{P}}$ via a one-hot vector $\boldsymbol{e}_k$ indicating the previous state $k$. Second, *recurrent feedback* governs the next term, via featurization functions for the system $g_\psi : \mathbb{R}^{DJ} \to \mathbb{R}^{\widetilde{R}}$ and for entities $f_\phi : \mathbb{R}^D \to \mathbb{R}^{\widetilde{D}}$ with parameters $\psi, \phi$ (known or learned) and weights $\boldsymbol{\Lambda}, \boldsymbol{\Psi}_j$.

Some applications may benefit from using available *exogenous covariates* to inform recurrent feedback, such as using the location of the ball itself or the remaining time on the game clock to better model future basketball player movement. Our general framework denotes such covariates at the system-level as $\boldsymbol{v}_{t-1}$ or entity-level as $\boldsymbol{u}_{t-1}^{(j)}$. When available, these covariates can drive transition probabilities; if no such variables exist they can be left out. Note that inference (Sec. 3) applies not merely to Equation 2.2, but to arbitrary instantiations.

**Emission model.** We generate the next observation for entity $j$ via a state-conditioned autoregression:

$$\boldsymbol{x}_t^{(j)} \mid \boldsymbol{x}_{t-1}^{(j)}, z_t^{(j)} \sim H_\zeta, \quad \text{where} \quad \zeta = \zeta(\boldsymbol{x}_{t-1}^{(j)}, z_t^{(j)}). \tag{2.3}$$

Users can select the emission distribution $H$ to match the domain of observed features $\boldsymbol{x}_t^{(j)}$: our later experiments use Gaussians for real-valued vectors and Von-Mises distributions for angles. The parameter $\zeta = \zeta(\boldsymbol{x}_{t-1}^{(j)}, z_t^{(j)})$ of the chosen $H$ depends on the previous observation $\boldsymbol{x}_{t-1}^{(j)}$ and current entity-level state $z_t^{(j)}$. We focus on lag 1 autoregression here, though extensions that condition on more than just one previous timestep are possible.

**Priors.** The Appendix describes prior distributions $p(\theta)$ on parameters assumed for the purpose of regularization. We use a "sticky" Dirichlet prior (Fox et al., 2011) to obtain smoother system-level segmentations.

**Specification.** To apply HSRDM to a concrete problem, a user must select the number of system states $L$ and entity states $K$ as well as functional forms of $g, f$. We assume that $g$ can be evaluated in $\mathcal{O}(J)$.

**Special cases.** If we remove the top-level system states $s_{0:T}$ (or equivalently set $L = 1$), our HSRDM reduces to separate models for each of $J$ entities, where each per-entity model is a recurrent autoregressive HMM (rAR-HMM) as in Linderman et al. (2017). If we remove the recurrence from our HSRDM, we obtain a multi-level autoregressive HMM that we refer to as a *hierarchical switching dynamical model* (HSDM). Later on, we compare our model to both ablations in several experiments.

## 3 Inference

Given observed time series $\boldsymbol{x}_{0:T}^{(1:J)}$, we now explain how to simultaneously estimate parameters $\theta$ and infer approximate posteriors over hidden states $s_{0:T}$ for the system and hidden states $z_{0:T}^{(1:J)}$ for all $J$ entities. Because all system-level and entity-level states are unobserved, the marginal likelihood $p(\boldsymbol{x}_{0:T} \mid \theta)$ is a natural objective for parameter estimation. However, exact computation of this quantity, by marginalizing over all

hidden states, is intractable. Given $L$ system-level states and $K$ entity-level states, computing $p(\boldsymbol{x}_{0:T} \mid \theta)$ naively via the sum rule requires a sum over $(LK^J)^T$ values. While the forward algorithm (Rabiner, 1989) resolves the exponential dependence in time, the exponential dependence in the number of entities persists: $TLK^{2J}$ operations are required to do forward-backward on HSRDMs. This exponential dependence remains prohibitively costly even in moderate settings; for instance, when $(T, J, L, K) = (100, 10, 2, 4)$, a direct application of the forward algorithm requires around 220 trillion operations.

Instead, we will pursue a structured approximation $q$ to the true (intractable) posterior over hidden states. Following previous work (Alameda-Pineda et al., 2021; Linderman et al., 2017), we define

$$q(s_{0:T}, z_{0:T}^{(1:J)}) = q(s_{0:T})\, q(z_{0:T}^{(1:J)}), \tag{3.1}$$

intending $q(s_{0:T}, z_{0:T}^{(1:J)}) \approx p(s_{0:T}, z_{0:T}^{(1:J)} \mid \boldsymbol{x}_{0:T}^{(1:J)}, \theta)$. Each factor of $q$ is parameterized separately, without dependence on $\theta$ or other random variables in the model. Each factor retains temporal dependency structure, avoiding the simplistic independence assumptions of complete mean-field inference (Barber et al., 2011).

Using this approximate posterior $q$, we can form a variational lower bound on the marginal log likelihood $\texttt{VLBO} \le \log p(\boldsymbol{x}_{0:T}^{(1:J)} \mid \theta)$, defined as $\texttt{VLBO}[\theta, q] = \mathbb{E}_q\big[\log p(\boldsymbol{x}_{0:T}^{(1:J)}, z_{0:T}^{(1:J)}, s_{0:T} \mid \theta)\big] + \mathbb{H}\big[q(z_{0:T}^{(1:J)}, s_{0:T})\big]$. As shown in the Appendix, computation of this bound scales as $O(TJL^2K^2)$, crucially *linear* rather than exponential in the number of entities $J$. This reduces the approximate number of operations required for inference on the earlier moderate example setting ($T{=}100, J{=}10, L{=}2, K{=}4$) from 220 trillion to 64 thousand.

To estimate $\theta$ and $q$ given data $\boldsymbol{x}_{0:T}$, we pursue coordinate ascent variational inference (CAVI; (Blei et al., 2017)) on the VLBO, known as variational expectation maximization (Beal, 2003) when $\theta$ is approximated with a point mass. Given a suitable initialization, we alternate between specialized update steps to each variational posterior or parameter:

$$q(s_{0:T}) \propto \exp\left\{\mathbb{E}_{q(z_{0:T}^{(1:J)})}[\log p(\boldsymbol{x}_{0:T}^{(1:J)}, z_{0:T}^{(1:J)}, s_{0:T} \mid \theta)]\right\}, \tag{3.2}$$

$$q(z_{0:T}^{(1:J)}) \propto \exp\left\{\mathbb{E}_{q(s_{0:T})}[\log p(\boldsymbol{x}_{0:T}^{(1:J)}, z_{0:T}^{(1:J)}, s_{0:T} \mid \theta)]\right\},$$

$$\theta = \arg\max_\theta\left\{\mathbb{E}_{q(z_{0:T}^{(1:J)})q(s_{0:T})}\left[\log p(\boldsymbol{x}_{0:T}^{(1:J)}, z_{0:T}^{(1:J)}, s_{0:T} \mid \theta)\right] + \log p(\theta)\right\}.$$

The updates above define the variational E-S step (VES step), variational E-Z step (VEZ step), and M-step, respectively. The first two formulas in equation 3.2 are derived by following the well-known generic variational recipe for optimal updates (Blei et al., 2017). The M-step allows the inclusion of an optional prior on some or all parameters. We've worked out efficient ways to achieve the optimal update for each step, as described below. Full details about each step, as well as recommendations for initialization, are in the Appendix. We also share code via the link at bottom of page 1, built upon JAX for efficient automatic differentiation (Bradbury et al., 2018).

**VES step for system-level state posteriors.** We can show the VES step reduces to updating the posterior of a surrogate Hidden Markov Model with $J$ independent autoregressive categorical emissions. Optimal variational parameters for this posterior can be computed via a dynamic-programming algorithm that extends classic forward-backward for an AR-HMM to handle recurrence. The runtime required is $\mathcal{O}\big(TJ(K^2 + KD + KL + KM) + TL^2\big)$.

**VEZ step for entity-level state posteriors.** We can show that the VEZ update reduces to updating the posterior of separate surrogate Hidden Markov Models for each entity $j$ with autoregressive categorical emissions which recurrently feedback into the transitions. Given a fixed system-level factor $q(s_{0:T})$, we can update the state posterior for entity $j$ independently of all other entities. This means inference is *linear* in the number of entities $J$, despite the fact that the HSRDM couples entities via the system-level sequence. The linearity arises even though our assumed mean-field variational family of Equation 3.1 did not make an outright assumption that $q(z_{0:T}^{(1:J)}) = \prod_{j=1}^{J} q(z_{0:T}^{(j)})$. Optimal variational parameters for this posterior can again be computed by dynamic programming that extends the forward-backward algorithm. The runtime required to update each entity's factor is $\mathcal{O}\big(T[K^2 + KD^2 + KL + KM]\big)$.

**M step for transition/emission parameters.** Updates to some parameters, particularly for emission model parameters when $H$ has exponential family structure (such as the Gaussian or Von-Mises AR likelihoods we use throughout experiments), can be done in closed-form. Otherwise, in general, we optimize $\theta$ by gradient ascent on the `VLBO` objective. This gradient ascent has the same per-iteration cost as the computation of the `VLBO`, with runtime $\mathcal{O}(TJL^2K^2)$. If the recurrence function parameters $\psi$ or $\phi$ are *learnable*, these can also be updated in the M step.

Like many variational methods, the alternating update algorithm in equation 3.2 provides a useful guarantee. Each successive step will improve the `VLBO` objective (or if incorporating priors, the modified objective $\mathtt{VLBO} + \log p(\theta)$) until convergence to a local maximum. This improvement assumes any M step that uses gradient ascent relies on a suitable implementation that guarantees a non-decrease in utility.

## 4 Related Work

Below we review several threads of the scientific literature in order to situate our work.

**Continuous representations of individual sequences.** Other efforts focus on latent continuous representations of individual time series. These can produce competitive predictions, but do not share our goal of providing a segmentation at the system and entity level into distinct and interpretable discrete regimes. Probabilistic models with continuous latent state representations are often based on classic linear dynamical system (LDS) models (Shumway & Stoffer, 1982). Deep generative models like the Deep Markov Model (Krishnan et al., 2017) and DeepState (Rangapuram et al., 2018) extend the LDS approach with more flexible transitions or emissions via neural networks.

**Discrete state representations of individual sequences.** Our focus is on discrete state representations which provide interpretable segmentations of available data, a line of work that started with classic approaches to entity-level-only sequence models like hidden Markov models (HMM) (Rabiner, 1989) and autoregressive hidden Markov models (AR-HMM) (Ghahramani & Hinton, 2000), later extended to include both discrete and continuous latent representations as in switching-state linear dynamical systems (SLDS) (Alameda-Pineda et al., 2021). Recent efforts such as DSARF (Farnoosh et al., 2021) and DS3M (Xu et al., 2025) have extended such base models to non-linear transitions and emissions via neural networks. All of these efforts still represent each time series with only entity-level (not system-level) dynamics, and do not incorporate recurrent feedback to guide the evolution of latent variables.

**Discrete states via recurrence on continuous observations.** Linderman et al. (2017) add a notion of *recurrence* to classic AR-HMM and SLDS models, increasing the flexibility in each timestep's transition distribution by allowing dependence on the previous continuous features, not just the previous discrete states. Later work has extended recurrence ideas in several directions that improve entity-level sequence modeling, such as multi-scale transition dependencies via the tree-structured construction of the TrSLDS (Nassar et al., 2019), recurrent transition models via SNLDS (Dong et al., 2020), or recurrent transition models that can explicitly model state durations via RED-SDS (Ansari et al., 2021). To model multiple recordings of worm neural activity, Linderman et al. (2019) pursue recurrent state space models that are described as *hierarchical* because they encourage similarity between each worm entity's custom dynamics model via common parameter priors in hierarchical Bayesian fashion. Their model assumes only entity-level discrete states.

**Multi-level discrete representations.** Stanculescu et al. (2014) developed a hierarchical switching linear dynamical system (HSLDS) for modeling the vital sign trajectories of individual infants in an intensive care unit. The root level of their directed graphical model assumes a discrete state sequence (analogous to our $s$) indicating whether disease was present or absent in the individual over time, while lower level discrete states (analogous to our $z$) indicate the occurrence of specific "factors" representing clinical events such as brachycardia or desaturation. While their graphical model also contains a multi-level discrete structure, we emphasize three key differences. First, they require *fully-supervised* data for training, where each timestep $t$ is *labeled* with top-level and factor-level states. In contrast, our structured VI routines to simultaneously estimate parameters and hidden states in the *unsupervised* setting are new. Second, their model does not incorporate recurrent feedback from continuous observations. Finally, they model individual time series not multiple interacting entities.

More recently, hierarchical time series models composed of Recurrent Neural Networks (RNNs) have been proposed for dynamical systems reconstitution (Brenner et al., 2025). Different from our work, this framework is not designed for modeling entity interactions within a single system, but modeling shared properties among multi-domain dynamical systems for time series transfer learning.

Lastly, Hierarchical Hidden Markov Models (HHMMs) (Fine et al., 1998) and their extensions (Bui et al., 2004; Heller et al., 2009) describe a single entity's observed sequence with multiple levels of hidden states. The chief motivation of the HHMM is to model different temporal length scales of dependency within an individual sequence. While HHMMs have been applied widely to applications like text analysis (Skounakis et al., 2003) or human behavior understanding (Nguyen et al., 2005), to our knowledge HHMMs have not been used to coordinate multiple entities overlapping in time.

**Models of teams in sports analytics.** Terner & Franks (2021) survey approaches to player-level and team-level models in basketball. Miller & Bornn (2017) apply topic models to tracking data to discover how low-level actions (e.g. run-to-basket) might co-occur among teammates during the same play. Metulini et al. (2018) model the convex hull formed by the court positions of the 5-player team throughout a possession via one system-level hidden Markov model. In contrast, our work provides a coordinated two-level segmentation representing the system as well as individuals.

**Personalized models.** Several switching state models assume each sequence in a collection have unique or personalized parameters, such custom transition probabilities or emission distributions (Severson et al., 2020; Alaa & van der Schaar, 2019; Fox et al., 2014). In this style of work, entity time series may be collected asynchronously, and entities are related by shared priors on their parameters. In contrast, we focus on entities that are synchronous in the same environment, and relate entities directly via a system-level discrete chain that modifies entity-level state transitions.

**Models of coordinated entities.** Several recent methods do jointly model multiple interacting entities or "agents", often using sophisticated neural architectures. Zhan et al. (2019) develop a variational RNN where trajectories are coordinated in short time intervals via entity-specific latent variables called "macro-intents". Yuan et al. (2021) develop the AgentFormer, a transformer-inspired stochastic multi-agent trajectory prediction model. Alcorn & Nguyen (2021) develop baller2vec++, a transformer specifically designed to capture correlations among basketball player trajectories. Xu et al. (2022) introduce GroupNet to capture pairwise and group-wise interactions. Unlike these approaches, ours builds upon switching-state models with *closed-form* posterior inference and produces *discrete* segmentations. Our approach may also be more sample efficient for applications with only a few minutes of data, as in Sec. 5.4.

**Models of interaction graphs.** Some works (Wu et al., 2020; Löwe et al., 2022) seek to learn an interaction graph from many entity-level time series. In such a graph, nodes correspond to entities and edge existence represents a direct, pairwise interaction between entities. Other works (Kipf et al., 2018; Webb et al., 2019) assume a fully-connected graph, but can learn to annotate each edge with a different *discrete type* representing different kinds of interaction. One possible type may be hard-coded to mean a "non-edge" for no interaction. Intentional priors can control the graph sparsity (frequency of non-edge labels). Recently, the *GRAph Switching dynamical Systems (GRASS)* approach (Liu et al., 2023) models interactions via a latent graph whose edges change dynamically over time.

For some applications, discovering possible pairwise interactions or interaction types that influence data is an explicit goal. In our chosen applications (e.g. basketball player movements), domain expertise indicates the graph is fully-connected. Moreover, when the graph is fully connected, interaction graph approaches burdensomely require runtimes that are quadratic in the number of entities $J$. In contrast, our approach models system-level dynamics explicitly with a more affordable runtime cost that is linear in $J$. Very recent work by Wang & Pang (2024) offers a different route to scaling to $J$ in the hundreds. Their approach can estimate directed graph structure by combining a variational dynamics encoder with partial correlation ideas.

# 5    Experiments

We now demonstrate our model's utility across several experiments. As we aim to highlight our model across two tasks - multi-step-ahead forecasting and interpretability of system dynamics - we show one of each on

synthetic and real datasets. In all but the first task, the data generation model is either unknown or not the same as the HSRDM. Within each task, we compare our model's performance to task-specific competitive baselines. We also compare to ablations that remove either the top-down influence of system-level hidden states or bottom-up recurrent feedback from observations. Overall, we see that our compact and efficient HSRDM is able to outperform alternatives with respect to discovering hidden system dynamics and maintain similar or better prediction performance than computationally intensive neural network methods.

### 5.1  FigureEight: Synthetic task of forecasting coordinated 2D trajectories

To illustrate the potential of our HSRDM for the purpose of **high-quality forecasting** of several coordinated entities, we study a synthetic dataset we call *FigureEight*. In the true generative process (detailed in App. D.3), each entity switches between clockwise motion around a top loop and counter-clockwise motion around a bottom loop. The overall observed entity-level 2D spatial trajectory over time approximates the shape of an "8" as in Fig. 2. Each entity has two true states, one for each loop, with a specific Gaussian vector autoregression process for each. Transition between these loop states depends on both top-down and bottom-up signals in the data-generating process. For top-down, a binary system-level state sets which loop is favored for all entities at the moment. For bottom-up, switches between loop states are only probable when the entity's current position is near the origin, where the loops intersect. Though coordinated, entity trajectories are not perfectly synchronized, varying due to individual rotation speeds and initial positions.

We generate data for 3 entities over 400 total timesteps. For training, we provide complete data for the first two entities (times 1-400) and partial data for the last entity (times 1-280). The prediction task of interest is *entity-specific partial forecasting*: estimate the remaining trajectory of entity $j = 3$ for times 281-400 (120 time steps), given partial information (timesteps 1-280) for that entity and full information (1-400) for other entities. The true trajectory for this heldout window is illustrated in Fig. 2: we see a smooth transition over time from the top loop to the bottom loop of the "8". We wish to compare our method to competitors at estimating this heldout trajectory.

**Baseline selection.**  To show the benefits of modeling system-level dynamics, we compare to baseline methods that can produce high-quality forecasts with discrete latent variables, yet only model entity-level (not system-level) dynamics and can do partial forecasts. First, we select the *deep switching autoregressive factorization model* (DSARF; (Farnoosh et al., 2021)). This model represents recent state-of-the-art forecasting performance and uses deep neural networks to flexibly define transition and emission structures yet still can produce discrete segmentations. Second, we compare to a *recurrent autoregressive hidden Markov model* (rAR-HMM; (Linderman et al., 2017)). This is an ablation of our method that removes our system-entity hierarchy. This experiment requires *partial forecasting* of one entity at times 281-400 given context from other entities. Some neural net baselines like Agentformer and GroupNet considered in later experiments do not easily handle such partial forecasting in released code, so we exclude them here on this task.

**Entity-to-system strategy.**  Since DSARF and rAR-HMM baselines each only model entity-level dynamics, for each one we try three different *entity-to-system strategies* to convert any entity-only model to handle a system of entities. First, complete *Independence* fits a separate model to each entity's data only, with no information flow between entities. Next, complete *Pooling* fits a single model on $N' = N * J$ total sequences, treating each sequence $n$ from each entity $j$ as an *i.i.d.* observation. Finally, *Concatenation* models a multivariate time series of expanded dimension $D' = J \cdot D$ constructed by stacking up all entity-specific feature vectors $\boldsymbol{x}_t^{(1)}, \boldsymbol{x}_t^{(2)}, \dots \boldsymbol{x}_t^{(J)}$ at each time $t$.

**Method configuration.**  For the HSRDM, we set $L = 2$ system states and $K = 2$ entity states. For emission model, we pick a Gaussian autoregressive to match the true process. We do not use any system-level recurrence $g$ as the true data generating process does not have this feedback. We set entity-level recurrence $f$ to a radial basis function indicating distance from the origin. While we do not expect this $f$ to improve training fit, we do expect it to improve forecasting, as it captures a key aspect of the true process: that switches between loops are only probable near the origin.

For all baselines (DSARF and rAR-HMM), we set the number of entity states at $K = 2$ to match the intended ground truth. The rAR-HMM uses the same emission model and entity-level recurrence $f$ as our HSRDM.

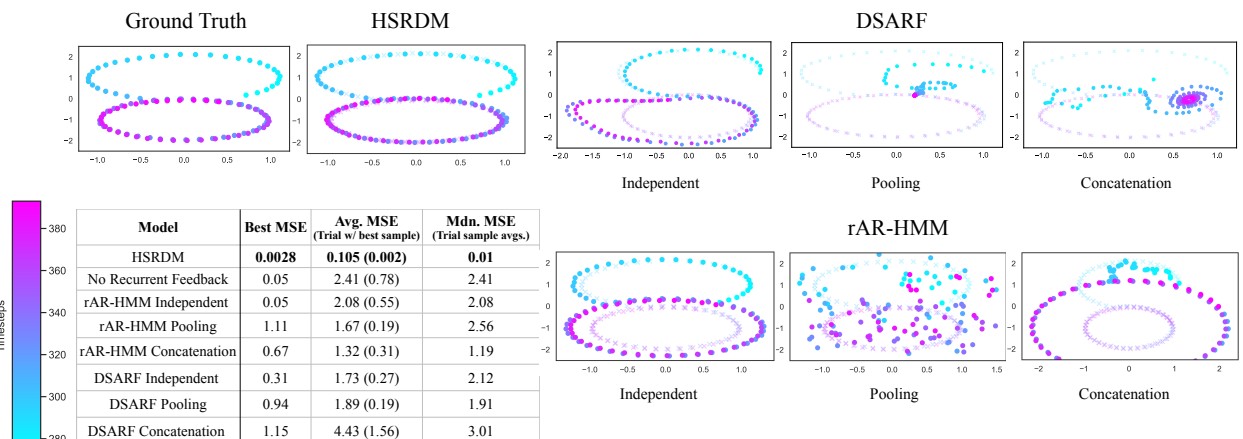

Figure 2: Predictions for the heldout time segment of one entity in *FigureEight* task. Models were trained on all data (1-400) from entities 1-2 and times 1-280 for entity 3, then asked to forecast the heldout period (times 281-400) for entity 3. Colors correspond to individual time steps, shown in the bar on the bottom left. X-marks in each panel show the ground truth trajectory; solid dots show the model forecast. **Top left**: Our best `HSRDM` prediction closely matches the ground truth at all time points. **Top right:** Best overall sample for the DSARF baseline under each strategy. **Bottom right:** Best overall sample for the rAR-HMM baseline, under each strategy (Indep., Pool, and Concat., defined in Sec. 5.1) for adapting an entity-only model to our hierarchical setting. rAR-HMM Independent is a "no system state" ablation of our model. **Bottom left:** Table reports for each model: best mean-squared-error (MSE) across independent trials and forecast samples, average MSE (with standard error in parens) across the samples from the trial with the best MSE, and median MSE across sample averages from all trials. This table includes ablation results for a version of our `HSRDM` with no recurrent feedback. Its forecasting plot can be found in App. D.3.

**Hyperparameter tuning.** Optimal hyperparameters for each model and entity-to-system strategy are determined independently. For `DSARF`, we tune its number of spatial factors and the lags indicating how the next timestep depends on the past. We don't tune any specific hyperparameters of our `HSRDM` or the `rAR-HMM`.

**Training.** For each tested method (where method means a model and (if needed) entity-to-system strategy), at each hyperparameter we train via 5 separate trials with different random seeds implying distinct initializations of parameters. This helps avoid local optima common to models with latent discrete state. `DSARF` is intentionally allowed more trials (10) to stress fair evaluation to external baselines. All `DSARF` models were trained with 500 epochs and a learning rate of 0.01. The `HSRDM` and its ablations are trained with 10 CAVI iterations. All models required similar training time on this small dataset: < 1 minute per run. Reproducible details for all methods (including specific hyperparameters) are in the Appendix and released code.

**Forecasting.** To forecast, from each trial's fit model we draw 5 samples of the heldout trajectory $x^3_{281:400}$ for target entity $j = 3$ over the time period of interest. We keep the "best" sample, meaning the sample with lowest mean squared error (MSE) compared to the true (withheld) trajectory. Each method can be represented by the trial and hyperparameter setting with lowest best-sample MSE, or via summary statistics across samples, trials, and hyperparameters.

**Results: Quantitative error.** For each method, we report the best-sample MSE in Fig. 2 (bottom left). Our `HSRDM` outperforms others by a wide margin, scoring a best-sample MSE less than 0.003 compared to the next-best value of 0.05 and values of 0.31 - 1.75 among others. To indicate reliability across forecast samples, we further report the average sample MSE from the trial that produces the best sample (with standard errors to indicate variation). This average is useful to distinguish between a trial that consistently produces quality forecasts from a trial that "gets lucky." While the `rAR-HMM` *Independent* model achieves a relatively low best MSE, the average MSE across all 5 samples from that trial is much worse than our method (2.08 vs. 0.105). Note that the `rAR-HMM` *Independent* model is a "no system state" ablation of the regular `HSRDM`. Our approach also outperforms a "no recurrent feedback" ablation (qualitative results in App. D.3).

To understand whether MSE is consistent across trials, we also report the median across trials of the average-sample MSE. For our `HSRDM`, the median-over-trials MSE is quite low (0.01), indicating that the `HSRDM` is typically reliable. The `HSRDM`'s best-sample trial happened to also yield one of the worst samples, which is why that trial's average-sample MSE is higher than the median. Comparing the best column to the median column, all methods show wide gaps indicating variation in quality across trials. There is a clear need to avoid local optima by considering many random initializations.

**Results: Visual quality.** Fig. 2 also shows the best sampled trajectory from each method. The forecast from our proposed `HSRDM` looks quite similar to the true trajectory. In contrast, every competitor struggles to reproduce the truth. The closest competitor method is the `rAR-HMM` with the Independent entity-to-system strategy. The main difference lies in how that model produced a larger outer circle for the bottom loop.

## 5.2 NBA Basketball: Real task of forecasting 2D player trajectories

We next aim to evaluate the `HSRDM`'s forecasting capabilities on real movements of professional basketball team. Specifically, we model the 5 players of the NBA's Cleveland Cavaliers (CLE), together with their 5 opponents, across multiple games in an open-access dataset (Linou, 2016) of player positions over time recorded from CLE's 2015-2016 championship season. To better evaluate the ability to model specific entities, we focused exclusively on 29 games involving one of CLE's four most common starting lineups. We randomly assigned these games to training (20 games), validation (4 games), and test (5 games) sets.

We split each game into non-overlapping basketball *event segments*, typically lasting 20 seconds to 3 minutes. Event segments contain periods of uninterrupted play (e.g. shot block $\rightarrow$ rebound offense $\rightarrow$ shot made) from the raw data, ending when there is an abrupt break in player motion or a sampling interval longer than the nominal sampling rate. Each event segment gives 2D court positions over time for all 10 players that form a multi-entity emission sequence $\boldsymbol{x}_{0:T}^{(1:J)}$. Each such sequence is modeled as an independent draw from our proposed `HSRDM` or competitor models. We standardized the court so that CLE's offense always faces the same direction (left), and downsampled the data to 5 Hz.

**Baseline selection.** We compare the forecasting performance of the `HSRDM` to multiple competitors. Because of `DSARF`'s poor prediction performance on even our synthetic task, here we instead explore `SNLDS` (Dong et al., 2020) as a neural network baseline which can provide a flexible model of complex time series. Because the `SNLDS` baseline is restricted to entity-level (not system-level) dynamics, we fit an independent `SNLDS` model to each player. To exemplify neural network methods that can predict trajectories of groups directly, we select `GroupNet` (Xu et al., 2022) and `Agentformer` (Yuan et al., 2021), given `GroupNet` and `Agentformer`'s leading reputations for high-quality forecasting of multiple entities (Xu et al., 2022; Yuan et al., 2021).

We further consider two ablations of our method. First, removing system-level states yields an independent `rAR-HMM` (Linderman et al., 2017) for each player. Second, we remove recurrent feedback. As in Yeh et al. (2019), we also try a simple but often competitive *fixed velocity* baseline.

**Model configuration.** The ground truth number of states is unknown; we pick $K = 10$ entity states and $L = 5$ system states. For our `HSRDM` and its ablations, our emissions distribution is a Gaussian vector autoregression with entity-state-dependent parameters (see Sec. E). System-level recurrence $g$ reports *all* player locations $\boldsymbol{x}_t^{(1:J)}$ to the system-level transition function, allowing future latent states to depend on player locations. Following Linderman et al. (2017), our entity-level recurrence function $f$ reports an individual player's location $\boldsymbol{x}_t^{(j)}$ and out-of-bounds indicators to that player's entity-level transition function, allowing each player's next state probability to vary over the 2D court. We use a sticky prior for system-level state transitions with $\alpha = 1$ and $\kappa = 50$ (see Sec. E.2 for details).

**Hyperparameters.** Optimal hyperparameters for each model and strategy are determined independently. For `GroupNet`, we used hyperparameters recommended by Xu et al. (2022) from their own basketball models. For `Agentformer`, we use the recommended architecture and training settings from Yuan et al. (2021), adjusting learning rates and number of epochs for our data (see App. E). For `SNLDS`, we use the architecture and training settings from Ansari et al. (2021) to model an electricity dataset, adjusting certain hyperparameters to match the `HSRDM` well (see App. E). We don't tune any hyperparameters of our `HSRDM` or its ablations. Ablations inherit hyperparameters like number of states where applicable.

Table 1: Quantitative evaluation on *Basketball*.

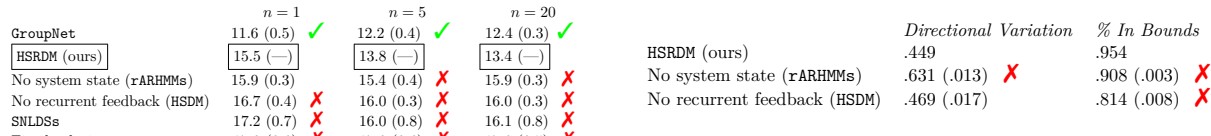

|  | $n = 1$ | | $n = 5$ | | $n = 20$ | |
|---|---|---|---|---|---|---|
| GroupNet | 11.6 (0.5) | ✔ | 12.2 (0.4) | ✔ | 12.4 (0.3) | ✔ |
| HSRDM (ours) | 15.5 (—) | | 13.8 (—) | | 13.4 (—) | |
| No system state (rARHMMs) | 15.9 (0.3) | | 15.4 (0.4) | ✘ | 15.9 (0.3) | ✘ |
| No recurrent feedback (HSDM) | 16.7 (0.4) | ✘ | 16.0 (0.3) | ✘ | 16.0 (0.3) | ✘ |
| SNLDSs | 17.2 (0.7) | ✘ | 16.0 (0.8) | ✘ | 16.1 (0.8) | ✘ |
| Fixed velocity | 17.0 (0.6) | ✘ | 17.0 (0.6) | ✘ | 17.0 (0.5) | ✘ |
| Agentformer | 33.1 (0.4) | ✘ | 21.3 (0.7) | ✘ | 25.9 (0.4) | ✘ |

(a) Forecast error (in feet) vs. train set size (num. games $n$).

|  | *Directional Variation* | | *% In Bounds* | |
|---|---|---|---|---|
| HSRDM (ours) | .449 | | .954 | |
| No system state (rARHMMs) | .631 (.013) | ✘ | .908 (.003) | ✘ |
| No recurrent feedback (HSDM) | .469 (.017) | | .814 (.008) | ✘ |

(b) Statistical comparisons to ablations.

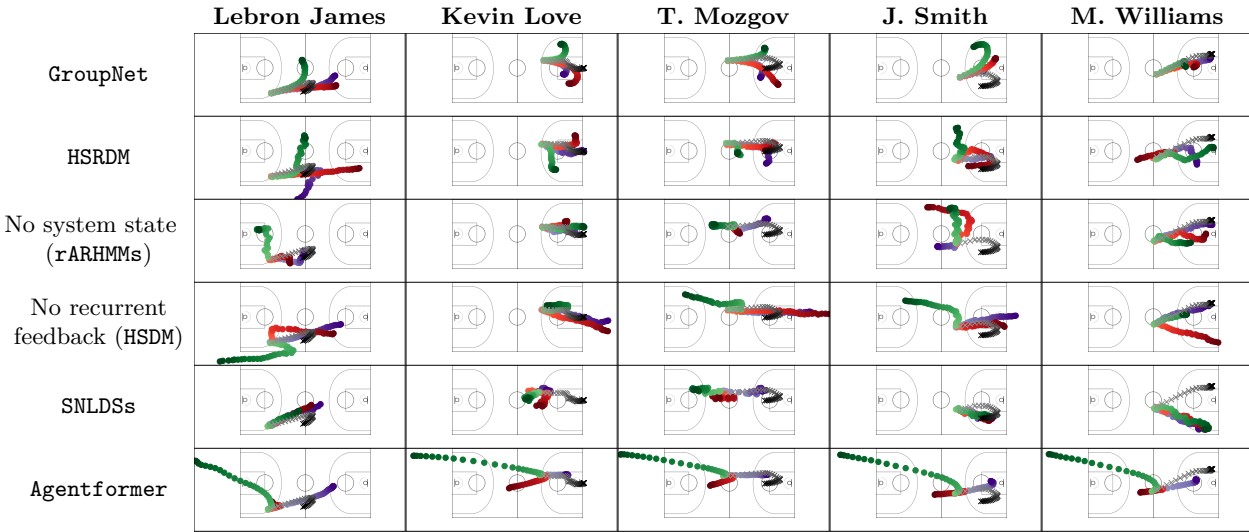

Figure 3: Sample forecasts of NBA player location trajectories. Grey tracks show true player trajectories from the 6-second forecasting window of the test event on which our model had median forecasting error. In color are three sampled forecasts from each model; purple/red/green indicate 1st/5th/10th best forecasting error (from 20 samples). Time runs from light to dark.

**Training time.** Our HSRDM is more computationally efficient for this task. Training an HSRDM on a 2023 Macbook with Apple M2 Pro chip on $n = 1, 5, 20$ training games took 2, 15, and 45 minutes, respectively. Training Agentformer on an Intel Xeon Gold 6226R CPU took 1.5, 6, and 13 *hours*, respectively. That is, our HSRDM was 17-45 times faster to train. We saw similar gains with the other neural network baselines; on the same 2023 Macbook, HSRDM trained 120 times faster than GroupNet and 10-50 times faster than SNLDS.

**Model size.** Our compact HSRDM has 9,930 parameters. In contrast, GroupNet has over 3.2 million parameters (320x larger) and Agentformer has over 6.5 million (650x larger). The collection of SNLDS models has about 0.5 million parameters (50x larger), even though this approach does not model cross-entity interactions.

**Evaluation procedure.** We randomly select a 6 second forecasting window within each of the 75 test set events. Preceding observations in the event are taken as context, and postceding observations are discarded. We sample 20 forecasts from each method for the 5 starting players on the Cavaliers. We report the mean distance in feet from forecasts to ground truth, with the mean taken over all events, samples, players, timesteps, and dimensions. We perform paired t-tests on the per-event differences in mean distances between our model vs competitors, using Benjamini & Hochberg (1995)'s correction for multiple comparisons over positively correlated tests. All tests were performed at the .05 significance level for two-sided tests.

**Results: Quantitative error.** Tab. 1a reports the mean distance in feet from forecasts to ground truth for each method. Methods whose forecasting error are significantly better (worse) than HSRDM according to the hypothesis tests are marked with a green check (red x). The standard error of the difference in means is given in parentheses. Our first key finding is that HSRDM provides better forecasts than ablations, supporting the utility of incorporating multi-level recurrent feedback and top-level system states into switching autoregressive models of collective behavior. Second, HSRDM provides better forecasts than some neural network baselines

(`SNLDSs` and `Agentformer`), as well as the fixed velocity baseline. Third, a neural network method designed to model coordinated entities (`GroupNet`) does provide the best overall forecasts in terms of error alone. Yet we consider our results "competitive" in that on a basketball court measuring 94x50 feet, our $n = 20$ forecasts are on average only one foot further off in terms of error while using a simple interpretable model that is roughly 320x smaller and 120x faster to train.

**Results: Qualitative forecasts.** Fig. 3 shows sampled forecasts from our model and baselines. As a first key finding, `HSRDM` forecasts are qualitatively more similar to `GroupNet` than the forecasts of other baselines. Second, system-level switches appear to help coordinate entities; players move in more coherent directions under `HSRDM` than the no system state ablation. Third, our multi-level recurrent feedback supports *location-dependent* state transitions; players are more likely to move to feasible *in-bounds* locations under `HSRDM` than without recurrence.

**Results: Statistical comparisons.** To corroborate the above conclusions from visual inspection, we examine two statistics on the entire test set: *Directional Variation*, which measures the coherence of movements by a basketball team via the variance across players of the movement direction on the unit circle between the first and last timesteps in the forecasting window, and *% In Bounds*, the mean percentage of each forecast that is in bounds. We report these statistics for the `HSRDM` and its ablations in Tab. 1b. Methods significantly different from the `HSRDM` baseline according to hypothesis testing are marked with a red "x." These results corroborate the intended purpose of each of our modeling contributions. Removing the system-level states significantly increases the directional variation across players, indicating lack of top-down coordination. Removing recurrence significantly reduces the percentage of forecasts that remain in bounds. Removing system states also significantly reduces the in bounds percentage, although to a lesser extent. We suggest that greater coordination in the player movement also helps keep the players in bounds.

### 5.3 MarchingBand: Synthetic task of interpreting marching band routines

Beyond the forecasting evaluations of earlier subsections, we now evaluate the `HSRDM`'s capabilities to discover useful and interpretable discrete system dynamics in systems with many ($J = 64$) entities. To do this, we introduce a synthetic dataset, *MarchingBand*, consisting of individual marching band players ("entities") moving across a 2D field in a coordinated routine to visually form a sequence of letters. Each observation is a position $\boldsymbol{x}_t^{(j)} \in \mathbb{R}^2$ of player $j$ at a time $t$ within the unit square centered at (0.5, 0.5) representing the field.

By design, player movement unfolds over time governed by both *top-down* and *bottom-up* signals. The team's goal is to spell out the word "LAUGH". An overall discrete system state sends the top-down signal of which particular letter, one of "L", "A", "U", "G", or "H", the players form on the field via their coordinated movements. Each entity has an assigned vertical position on the field (when viewed from above), and moves horizontally back and forth to "fill in" the shape of the current letter. Example frames are shown in Fig. 4. Each state is stable for 200 timesteps before transitioning in order to the next state.

Each entity's position over time on the unit square follows the current letter's top-down prescribed movement pattern perturbed by small-scale i.i.d. zero-mean Gaussian noise. When reaching a field boundary, typically the player is reflected back in bounds instantaneously. However, there is a small chance an entity will continue out-of-bounds (OOB, $\boldsymbol{x}_{t1} \notin (0, 1)$). When enough players go OOB, this bottom-up signal triggers the system state immediately to a special "come together and reset" state, denoted "C". This reset state does not form the letter "C". Instead, in state "C" all players move to the center for the next 50 timesteps, then return to repeat the most recent letter before continuing on to remaining letters.

Altogether, we build a dataset of $N = 10$ independently sampled sequences of "LAUGH" each with $J = 64$ marching band players. Each sequence contains a different number of time-steps, ranging from 1000 to 1100, depending on how many reset segments occur. To trigger reset state "C", we use a threshold of 11 players OOB as this generates a moderate number of 6 "C" segments distributed across the 10 sequences.

**Research goal.** We seek to understand whether our proposed model or competitors can discover the overall discrete system-level dynamics, as measured by discrete segmentation quality. The true data-generating process, while similar to the `HSRDM` in using top-down and bottom-up signals, is not strictly an `HSRDM` generative model because each state has fixed duration. Thus, all methods are somewhat misspecified here.

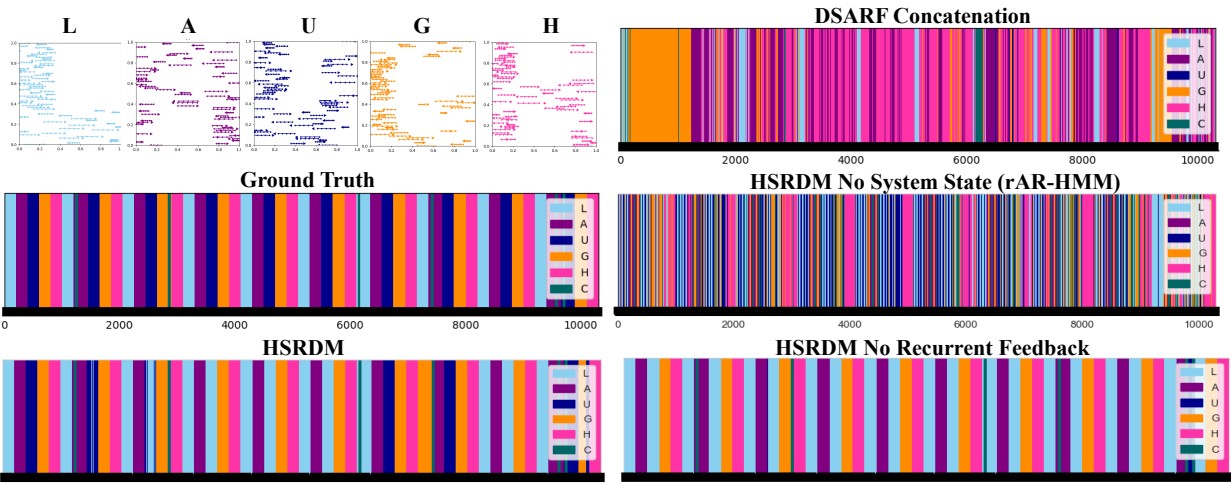

Figure 4: System-level segments for all models on the *MarchingBand* task. **Top left**: An example snapshot in time of ground truth entity observations for each system-state letter (spells "LAUGH"). **Bottom left**: The ground truth segmentation across all 10 sequences as well as the `HSRDM` segmentation. **Right side**: Results for ablations (no system state (rAR-HMM), no recurrent feedback) as well as DSARF.

**Baseline selection.** We focus on methods that can produce an estimated discrete segmentation at the system-level. Using this criteria, our primary external competitor is `DSARF` (Farnoosh et al., 2021). Neither `GroupNet` nor `Agentformer` can produce a segmentation or offer interpretable discrete system states.

Additionally, we compare to two ablations of our method: removing the system-level state (leaving an `rAR-HMM`) and removing the recurrent feedback (leaving a multi-level HMM). For both `DSARF` and `rAR-HMM`, we obtain system-level segmentations in two steps. First, we use the *Independent* strategy ( Sec. 5.1) to obtain entity-level segmentations. Next, for each timestep we concatenate the one-hot indicator vectors of each entity to form a longer vector of size $J \times K$. These per-time features are clustered via k-means to obtain a system-level segmentation. This should favor clustering timesteps that share entity state assignments.

**Model configuration.** For all methods, we fairly provide knowledge of the true number of system states, $6 = |\{$L, A, U, G, H, C$\}|$. Models with system states (`HSRDM` and its recurrent ablation) have $L = 6$ system states and $K = 4$ entity states. Models with only entity-level states (`DSARF` and `rAR-HMM`) are fit with $K = 6$ entity states, imagining one state per letter, followed by k-means to discover 6 system-level states. For `HSRDM`, we set system-level recurrence $g$ to count the number of entities out of bounds. For `HSRDM` and `rAR-HMM`, we set entity-level recurrence $f$ to the identity function. The emission model is set to a Gaussian Autoregressive.

**Hyperparameter tuning.** For `DSARF`, we again select the number of spatial factors and the lags, via grid search (see Appendix F.4). Ultimately, we select 25 spatial factors and the set of lags $\{1, ..., 200\}$, suggesting long-range dependency is useful to compensate for lack of recurrence. For our `HSRDM` and ablations, no specific hyperparameter search was done. Methods were trained with similar epochs/iterations as the *FigureEight* data, with reasonable convergence verified by trace plots. To avoid local optima, for each method we take the best trial (in terms of segmentation accuracy) of 5 possible seeds controlling initialization.

**Results: segmentation quality.** Fig. 4 allows visual comparison of the ground truth system-level segmentation and the estimated segmentations from various methods. Estimated states are aligned to truth by minimizing classification accuracy (Hamming distance of one-hot indicators) via the Munkres algorithm (Munkres, 1957). The visualized segmentations depict each model's best classification accuracy

Table 2: Accuracy of system-level segmentation on *Marching-Band*, compared to ground-truth after alignment.

| Method | Best Acc. | Median Acc. |
|---|---|---|
| HSRDM | 88% | 83% |
| No Recur. Feedback (HSDM) | 80% | 69% |
| No System State (rAR-HMM) | 48% | 42% |
| DSARF | 38% | 32% |

over multiple training trials with different random initializations. `DSARF` was given 10 chances and all others 5 in an attempt to provide `DSARF` with a more-than-fair chance. As reported in Tab. 2, our `HSRDM` obtains 88% accuracy overall, with clear recovery of each of the 6 states in visuals. In contrast, other methods struggle, delivering $38 - 80\%$ accuracy and notably worse segmentations in Fig. 4. Even with respect to median accuracy across trials (also in Tab. 2), the `HSRDM` outperforms its competitors. Inspecting Fig. 4, the `HSRDM` recovers each true state most of the time. In contrast, the next-best method, the no recurrence ablation, completely misses capturing the "U" system-level state across examples even in its best trial. This experiment highlights the `HSRDM` as a natural model to recover hidden system dynamics when many individuals influence collective behavior. We provide supplemental results for accurate estimation of system states for a larger number of entities ($J = 200$) and a larger number of dimensions ($D = \{10, 30\}$) in the Appendix F.3.

### 5.4 Soldier Training: Real task of interpreting risk mitigation strategies

For our final experiment, we model a squad of active-duty U.S. Army soldiers during a training exercise designed to improve team coordination. Our goal is to demonstrate the `HSRDM`'s capability to reveal useful and interpretable group dynamics in a real application. For simplicity and to highlight our model's capability as an explanatory tool, we focus on interpreting one model's fit rather than comparisons to alternatives.

In this training exercise, the squad's task is to maintain visual security of their entire perimeter while simulated enemy fire comes primarily from the south. Focus on the south creates a potential blindside to the north. If this blindside is left unchecked for a sufficiently long time, this leaves the squad vulnerable to a blindside attack. Among several goals, the squad was instructed that a primary goal was mitigating overall risk with visual security as a key sub-task. As a way to mitigate risk, the squad should periodically plan to have at least one soldier briefly turn their head to the north to regain visibility and reduce vulnerability to a blindside attack. Strong performance at this sub-task requires coordination across all soldiers in the squad. Our modeling aim is to utilize the `HSRDM` to interpret the soldiers' risk mitigation strategy with respect to checking their blindside as they complete the overall task.

The data consists of univariate time series of heading direction angles $x_t^{(j)}$ recorded at 130 Hz from each soldier's helmet inertial measurement unit (IMU), downsampled to 6.5 Hz. We have one 12 minute recording of one squad of 8 soldiers. Raw data from the first minute of contact is illustrated in Fig. 5. Due to privacy concerns, the dataset is not shareable. This study was approved by the U.S. Army Combat Capabilities Development Command Armaments Center Institutional Review Board and the Army Human Research Protections Office (Protocol #18-003).

**Model configuration.** Our `HSRDM` captures the risk mitigation strategies of the group by setting the system-level recurrence function $g$ to the normalized elapsed time since any one of the $J$ soldiers looked within the north quadrant of the circle. No entity-level recurrence $f$ was included. Soldier headings must remain on the unit circle throughout time, so we use a *Von Mises autoregression* as the emission model $H_\zeta$ for the $k$-th state of the $j$-th soldier:

$$x_t^{(j)} \mid x_{t-1}^{(j)}, \{z_t^{(j)} = k\} \sim \mathcal{VM}\left(\mu_{j,k}\big(x_{t-1}^{(j)}\big) \, , \, \kappa_{j,k}\right), \text{where } \mu_{j,k}(x_{t-1}^{(j)}) = \alpha_{j,k} \, x_{t-1}^{(j)} + \delta_{j,k}. \tag{5.1}$$

The Von Mises distribution (Banerjee et al., 2005; Fisher & Lee, 1994), denoted $\mathcal{VM}(\mu, \kappa)$, is a exponential family distribution over angles on the unit circle, governed by mean $\mu$ and concentration $\kappa > 0$. Here, $\alpha_{j,k}$ is an autoregressive coefficient, $\delta_{j,k}$ is a drift term, and $\kappa_{j,k}$ is a concentration for entity $j$ in state $k$.

We set the number of entity- and system-level states to $K = 4$ and $L = 3$, based on a quick exploratory analysis. We use a sticky Dirichlet prior for system-level transitions, as in Equation B.2, with $\alpha = 1.0$ and $\kappa = 50.0$, so that the prior puts most probability mass on self-transition probabilities between .90 and .99.

**Training.** We applied the CAVI inference from Sec. 3, observing suitable convergence after 10 iterations.

**Results.** Fig. 5 visualizes key results. Inspection of the inferred system-level states, entity-level states, and learned transition probabilities suggests that the model learns a special risk mitigation strategy for Soldier 6. Specifically, the model learns a "turn north" state (blue) for Soldier 6 that is particularly probable when the entire squad reaches a system-level "high elapsed time since any blindside check" state (red).

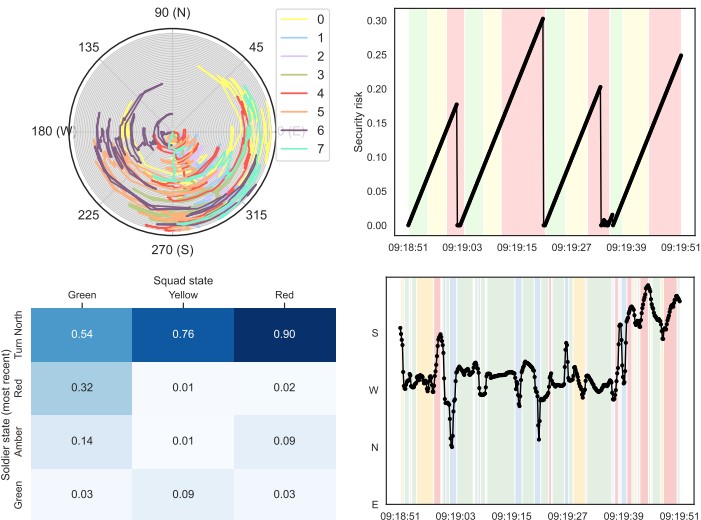

Figure 5: *Modeling the heading directions of a squad of soldiers engaged in simulated battle.* **Top left:** Heading directions (in degrees) of a squad of soldiers over time. Each color represents a different soldier. Time moves from center of circle to boundary. **Top right:** Inferred squad-level states $s_{0:T}$ (colors) superimposed over black curve representing the squad's cumulative security risk in the north direction (elapsed time since any soldier checked their blindside) as a function of time. The learned red squad state seems to indicate high security risk. These squad states can modulate soldier-level heading dynamics. **Bottom right:** Inferred entity-level states $z_{0:T}$ (colors) for Soldier 6, superimposed on observed time series of heading direction from that soldier's helmet IMU. The light blue state's autoregressive emission dynamics produce a rapid turn to the north. Twice this state persisted long enough for the soldier to reduce security risk in the north (around 19:03 and 19:20). **Bottom left:** The learned probability that Soldier 6 turns to the north from various soldier-specific states ($z$, rows) depends upon the squad-level states ($s$, columns). The soldier is most likely to persist in turning north when the squad has a security vulnerability ($s$ is red).

## 6   Discussion & Conclusion

We have introduced a cost-efficient family of models for capturing the dynamics of individual entities evolving in coordinated fashion within a shared environment over the same time period. These models admit efficient structured variational inference in which coordinate ascent can alternate between E-step dynamic programming routines similar to classic forward-backward recursions to infer hidden state posteriors at both system- and entity-levels and M-step updates to transition and emission parameters that also use closed-form updates when possible. Across several datasets, we have shown our approach represents a natural way to capture both top-down system-to-entity and bottom-up entity-to-system coordination while keeping costs linear in the number of entities.

**Limitations.** Our method intentionally prioritizes discrete latent variables for interpretability and efficiency. For some tasks, this choice may be less flexible than more rich continuous representations. We further assume interactions between entities are moderated by a top-level discrete variable. This choice allows runtime to be linear in the number of entities $J$, but may be less expressive than direct pairwise interaction. Several coordinate ascent steps in any per-entity `rAR-HMM` with Gaussian emissions scale quadratically in $D$ due to our choice to parameterize via a full covariance matrix. Scaling beyond a few dozen features may require diagonal or low-rank parameterizations. Furthermore, the parametric forms of both transitions and emissions in our model allow tractability but clearly limit expressivity compared to deep probabilistic models that integrate non-linear neural nets (Krishnan et al., 2017). Scaling to many more entities would require extensions of our structured VI to process minibatches of entities (Hoffman et al., 2013; Hughes et al., 2015). Scaling to much longer sequences might require processing randomly sampled windows (Foti et al., 2014).

**When to favor `HSRDM` over alternatives?** Readers may wonder when to expect our `HSRDM` has advantages over more flexible non-linear models. We suggest that our approach may be favorable when there is clear

*top-down* coordination among 5-200 entities that could be expressed via simple discrete segmentations by a domain expert, and there is interest in both the forecasting and explanatory purposes of the model. When bottom-up coordination alone dominates, interaction graph approaches or neural net methods like GroupNet or AgentFormer may be better. Our basketball case study highlights this: GroupNet slightly outperforms our model in forecasting accuracy. We think this is in part because basketball play is driven by dynamically responding to in-moment conditions (pairwise interactions) rather than rigidly following the prescribed plays called by a coach. Our parameter-efficient approach also has advantages when number of available time series is relatively limited (say a few hundred recordings or less). When abundant data exists, flexible function approximation methods for forecasting may be preferred.

**Future directions.** For some applied tasks, it may be promising to extend our two-level system-entity hierarchy to even more levels (e.g. to represent nested structures of platoons, squads, and individual soldiers all pursing the same mission). Additionally, we could extend from `rARHMMs` to switching linear dynamical systems by adding an additional latent continuous variable sequence between discretes $z$ and observations $x$ in the graphical model, or avoid first-order Markov dependency in generating state sequences via more flexible distributions parameterized by recurrent neural nets. Lastly, we can add selective and adaptive recurrence functions for modeling entity interactions where current observations influence the type of signal that would be useful feedback for the hidden states.

## Broader Impacts

Our work attempts to provide a fundamental framework for modeling interacting entities via explicit mechanisms for top-down and bottom-up influences on group dynamics. As with many technological innovations, for specific applications there may be positive and negative downstream consequences. It is important for researchers to take an active role to avoid misuse and harm.

Many possible applications involve human subjects. We strongly recommend all researchers follow appropriate local regulations and best practices for human subjects research. Each application must carefully consider ethical principles such as respect for persons and beneficence (maximizing benefits while minimizing risks). When using this model for human subjects research, please use care when making specific decisions about what covariates to include. Consider de-identification whenever possible to preserve the privacy of individuals.

Some potential applications of our work involve modeling human teams working in military applications. Our present work focuses exclusively on deidentified data from *simulated* training exercises via a research plan that was approved by a U.S. Army institutional review board. We strongly recommend similar ethics oversight and care in future use cases.

## Acknowledgments

This research was sponsored in part by the U.S. Army DEVCOM Soldier Center, and was accomplished under Cooperative Agreement Number W911QY-19-2-0003. The views and conclusions contained in this document are those of the authors and should not be interpreted as representing the official policies, either expressed or implied, of the U.S. Army DEVCOM Soldier Center, or the U.S. Government. The U. S. Government is authorized to reproduce and distribute reprints for Government purposes notwithstanding any copyright notation hereon.

Authors KG, EM, and MCH also acknowledge funding from the U. S. National Science Foundation under Growing Convergence Research grant #2428640. MCH is further supported by NSF CAREER award #2338962 and a gift from Apple, Inc.

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

# Contents

## Code

Source code for running our proposed `HSRDM` and reproducing experiments in this paper can be found at
https://github.com/tufts-ml/team-dynamics-time-series/

## A  ARHMM Method Details

As detailed below in Sec. A.1, a recurrent autoregressive Hidden Markov Model (`rAR-HMM`) generalizes a standard Hidden Markov Model by adding autoregressive and recurrent edges to the probabilistic graphical model. Although `rAR-HMM` models have been previously proposed in the literature (Linderman et al., 2017), we do not know of any explicit proposition (or justification) describing how to perform posterior state inference for these models. The literature provides such a proposition for (non-recurrent) autoregressive Hidden Markov Model (`ARHMMs`; e.g., see (Hamilton, 1994)), but not for `rARHMMs`. Hence, we provide the missing propositions with proofs here; see Props. A.2.1 and A.2.2. We believe that these explicit propositions can be useful when composing recurrence into more complicated constructions. Indeed, we use them throughout the supplement in order to derive inference for our `HSRDMs`; for example, see the VES step in Sec. B.2 or the VEZ step in Sec. B.3. In fact, we also utilize the proofs of these propositions when describing how to perform inference with `HSRDMs` when the dataset is partitioned into multiple examples; see Sec. B.7.

### A.1  Model

The complete data likelihood for a $(K, m, n)$-order recurrent AR-HMM (`rAR-HMM`) is given by Radon-Nikodỳm density

$$p(\boldsymbol{x}_{1:T}, \boldsymbol{z}_{1:T} \mid \theta) = \underbrace{p(z_1 \mid \theta)p(\boldsymbol{x}_1 \mid z_1, \theta)}_{\text{initialization}} \prod_{t=2}^{T} \underbrace{p(z_t \mid z_{t-1}, \boldsymbol{x}_{(t-m):(t-1)}, \theta)}_{\text{transitions}} \underbrace{p(\boldsymbol{x}_t \mid z_t, \boldsymbol{x}_{(t-n):(t-1)}, \theta)}_{\text{emissions}} \tag{A.1}$$

where $\boldsymbol{x}_{1:T}$ are the observations, $\boldsymbol{z}_{1:T} \in \{1, \dots, K\}$ are the discrete latent states, and $\theta$ are the parameters. The `rAR-HMM` generalizes the standard HMM (Rabiner, 1989), which contains neither autoregressive emissions (blue) nor recurrent feedback (red) from emissions to states. The $(K, m, n)$-order `rAR-HMM` gives a $(K, n)$-order autoregressive HMM (`ARHMM`) in the special case where

$$p(z_t \mid z_{t-1}, \boldsymbol{x}_{(t-m):(t-1)}, \theta) = p(z_t \mid z_{t-1}, \theta) \tag{A.2}$$

See Fig. A.1 for a probabilistic graphical model representation in the special case of first-order recurrence ($m = 1$) and autoregression ($n = 1$).

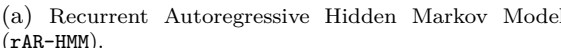

(a) Recurrent Autoregressive Hidden Markov Model (`rAR-HMM`).

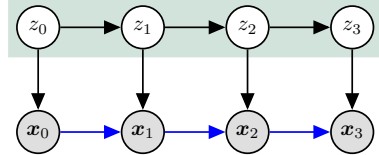

(b) Autoregressive Hidden Markov Model (`ARHMM`).

Figure A.1: Probabilistic graphical model representation of a Recurrent Autoregressive HMM (`rAR-HMM`), and its special case, an Autoregressive HMM (`ARHMM`). For simplicity the illustration assumes first-order autoregression and recurrence, but higher-order dependencies can also be accomodated (see Equation A.1 and Props. A.2.1 and A.2.2). Autoregressive edges are shown in blue and recurrent edges are shown in red.

**Remark A.1.1.** (*On generalizing a `HMM` with autoregressive emissions and recurrent state transitions.*) Let us highlight how a `rAR-HMM` model generalizes a conventional `HMM`:

- *Recurrence*: A lookback window of $n$ previous observations $\boldsymbol{x}_{t-m:t-1}$ can influence the transitions structure for the current state $z_t$.

$$\mathbf{pa}(z_t) = \{z_{t-1}\} \cup \underbrace{\{\boldsymbol{x}_{(t-m):(t-1)}\}}_{\text{if recurrent}}$$

- *Autoregression*: A lookback window of $m$ previous observations $\boldsymbol{x}_{t-n:t-1}$ can influence the emissions structure for the current observation $\boldsymbol{x}_t$.

$$\mathbf{pa}(\boldsymbol{x}_t) = \{z_t\} \cup \underbrace{\{\boldsymbol{x}_{(t-n):(t-1)}\}}_{\text{if autoregressive}}$$

Note in particular that each node (observation $\boldsymbol{x}_t$ or state $z_t$) can have *many* parents among previous observation variables $\boldsymbol{x}_{1:t}$, but only *one* parent among state variables $z_{1:t}$ (namely, the closest in time from the present or past).[1] This assumption will be important when deriving the smoother in Sec. A.2. △

## A.2 State Estimation

Here we discuss state estimation for the `rAR-HMM`. We begin with some notation.

**Notation A.2.1.** Given a sequence of observations up to some time $t$, we can define the conditional probability of the state $z_s$ at a target time $s \in \{1, 2, \ldots T\}$ via the probability vector $\boldsymbol{\xi}_{s \mid t} \in \Delta_{K-1} \subset \mathbb{R}^K$. The $k$-th element of this vector is given by $p_\theta(z_s = k \mid \boldsymbol{x}_{1:t})$. That is,

$$\boldsymbol{\xi}_{s \mid t} \triangleq p_\theta(z_s \mid \boldsymbol{x}_{1:t}) = \left[ p_\theta(z_s = 1 \mid \boldsymbol{x}_{1:t}), \ldots, p_\theta(z_s = K \mid \boldsymbol{x}_{1:t}) \right]^T$$

Using this notation, we can define three common inferential tasks:

1. *Filtering.* Infer the current state given observations $\boldsymbol{\xi}_{t \mid t} = p_\theta(z_t \mid \boldsymbol{x}_{1:t})$.

2. *Smoothing.* Infer a past state given observations $\boldsymbol{\xi}_{s \mid t} = p_\theta(z_s \mid \boldsymbol{x}_{1:t})$, where $s < t$.

3. *Prediction.* Predict a future state given observations, $\boldsymbol{\xi}_{u \mid t} = p_\theta(z_u \mid \boldsymbol{x}_{1:t})$, where $u > t$.

△

Now we can give Props. A.2.1 and A.2.2, which parallel the presentation of the Kalman filter and smoother in the context of state space models (Shumway & Stoffer, 2000; Hamilton, 1994). In particular, we will present the forward algorithm in terms of a *measurement update* (which uses the observation $\boldsymbol{x}_t$ to transform

---

[1]What if we wanted to relax the specification so that the emissions could depend on a finite number $M$ of previous states $p(\boldsymbol{x}_t \mid z_t, z_{t-1}, \ldots, z_{t-M}, \boldsymbol{x}_{1:t-1}, \phi)$? This situation can be handled by simply redefining the states in terms of tuples $z_t^* = (z_t, z_{t-1}, \ldots z_{t-M})$, such that $z_t^*$ takes on $K^M$ possible values, one for each sequence in the look-back window (Hamilton, 2010, pp.8).

$\boldsymbol{\xi}_{t \mid t-1}$ into $\boldsymbol{\xi}_{t \mid t}$) and a *time update* (which transforms $\boldsymbol{\xi}_{t \mid t}$ into $\boldsymbol{\xi}_{t+1 \mid t}$, without requiring an observation). These propositions show that *filtering* and *smoothing* can be done using the same recursions as used in a classical HMM (Hamilton, 1994), except that the variable interpretations differ for both the emissions step and transition step. In the statements and proofs below, we continue to use the same color scheme as was used in Equation A.1 and Fig. A.1, whereby blue designates autoregressive edges and red designates recurrence edges in the graphical model. These colors highlight differences from classic HMMs, which lack both types of edges.

**Proposition A.2.1. (Filtering a Recurrent Autoregressive HMM.)** *Filtered probabilities* $\boldsymbol{\xi}_{t \mid t} \triangleq p_\theta(z_t \mid \boldsymbol{x}_{1:t})$ *for a Recurrent Autoregressive Hidden Markov Model can be obtained by recursively updating some initialization* $\boldsymbol{\xi}_{1 \mid 0}$ *by*

- Measurement update.

$$\boldsymbol{\xi}_{t \mid t} = \frac{(\boldsymbol{\xi}_{t \mid t-1} \odot \boldsymbol{\epsilon}_t)}{\mathbf{1}^T(\boldsymbol{\xi}_{t \mid t-1} \odot \boldsymbol{\epsilon}_t)}$$

- Time update.

$$\boldsymbol{\xi}_{t+1 \mid t} = \boldsymbol{A}_t \, \boldsymbol{\xi}_{t \mid t}$$

*where here* $\boldsymbol{\epsilon}_t = (\epsilon_{t1}, \dots \epsilon_{tk}) = (p_\theta(\boldsymbol{x}_t \mid z_t = k, \boldsymbol{x}_{1:t-1}))_{k=1}^K$ *is the* $(K \times 1)$ *vector whose k-th element is the emissions density,* $\boldsymbol{A}_t$ *represents the* $(K \times K)$ *transition matrix whose* $(k, k')$*-th element is* $p_\theta(z_{t+1} = k' \mid z_t = k, \boldsymbol{x}_{1:t})$, $\mathbf{1}$ *represents a* $(K \times 1)$ *vector of 1s, and the symbol* $\odot$ *denotes element-by-element multiplication (Hadamard product).*

*Proof.*

- Measurement update.

$$\begin{aligned}
\boldsymbol{\xi}_{t \mid t} &= p_\theta(z_t \mid \boldsymbol{x}_{1:t}) && \text{Notation} \\
&\propto p_\theta(z_t, \boldsymbol{x}_t \mid \boldsymbol{x}_{1:t-1}) && \text{Conditional density} \\
&= \underbrace{p_\theta(z_t \mid \boldsymbol{x}_{1:t-1})}_{\triangleq \, \boldsymbol{\xi}_{t \mid t-1}} \odot \underbrace{p_\theta(\boldsymbol{x}_t \mid z_t, \boldsymbol{x}_{1:t-1})}_{\triangleq \, \boldsymbol{\epsilon}_t} && \text{Chain rule of probability} \\
\implies \boldsymbol{\xi}_{t \mid t} &= \frac{(\boldsymbol{\xi}_{t \mid t-1} \odot \boldsymbol{\epsilon}_t)}{\mathbf{1}^T(\boldsymbol{\xi}_{t \mid t-1} \odot \boldsymbol{\epsilon}_t)} && \text{Normalize}
\end{aligned}$$

- Time update.

$$\begin{aligned}
\boldsymbol{\xi}_{t+1 \mid t} &= p_\theta(z_{t+1} \mid \boldsymbol{x}_{1:t}) && \text{Def.} \\
&= \sum_{k=1}^K p_\theta(z_{t+1}, z_t = k \mid \boldsymbol{x}_{1:t}) && \text{Law of Total Prob.} \\
&= \sum_{k=1}^K p_\theta(z_{t+1} \mid z_t = k, \boldsymbol{x}_{1:t}) \, p_\theta(z_t = k \mid \boldsymbol{x}_{1:t}) && \text{Chain rule of probability} \\
&= \sum_{k=1}^K \underbrace{\left[\boldsymbol{A}_t\right]_{k,:}}_{k\text{th row of } \boldsymbol{A}_t} \underbrace{[\boldsymbol{\xi}_{t \mid t}]_k}_{k\text{th element of } \boldsymbol{\xi}_{t \mid t}} && \text{Notation} \\
&= \boldsymbol{A}_t \, \boldsymbol{\xi}_{t \mid t} && \text{Def. matrix multiplication}
\end{aligned}$$

$\square$

**Remark A.2.1.** (*Initializing the filtering algorithm in Prop. A.2.1.*) Inspired by Hamilton (1994, pp.693), we provide some suggestions for initializing the filtering algorithm of Prop A.2.1. In particular, we can set $\boldsymbol{\xi}_{1 \mid 0}$ to

- Any reasonable probability vector, such as the uniform distribution $K^{-1}\mathbf{1}$.

- The maximum likelihood estimate.

- The steady state transition probabilities, if they exist.

$\triangle$

**Proposition A.2.2. (Smoothing a Recurrent Autoregressive Hidden Markov Model.)** *Smoothed probabilities* $\boldsymbol{\xi}_{t\,|\,T} \triangleq p_\theta(z_t \mid \boldsymbol{x}_{1:T})$ *for a Hidden Markov Model can be obtained by the recursion*

$$\boldsymbol{\xi}_{t\,|\,T} = \boldsymbol{\xi}_{t\,|\,t} \odot \left\{ \boldsymbol{A}_t^T \cdot \left[ \boldsymbol{\xi}_{t+1\,|\,T} \, (\div) \, \boldsymbol{\xi}_{t+1\,|\,t} \right] \right\}$$

*where the formula is initialized by* $\boldsymbol{\xi}_{T\,|\,T}$ *(obtained from the filtering algorithm of Prop. A.2.1) and is then iterated backwards for* $t = T-1, T-2, \ldots, 1$, *in a step analogous to the backward pass of the classic forward-backward recursions for plain HMMs (Rabiner, 1989). Here,* $\boldsymbol{A}_t$ *represents the* $(K \times K)$ *transition matrix whose* $(k, k')$-*th element is* $p_\theta(z_{t+1} = k' \mid z_t = k, \boldsymbol{x}_{1:t})$, *the symbol* $\odot$ *denotes element-wise multiplication, and the symbol* $(\div)$ *denotes element-wise division.*

*Proof.* [2]

We proceed in steps:

- ⬛ Step 1 We show $\boxed{p_\theta(z_t \mid z_{t+1}, \boldsymbol{x}_{1:T}) = p_\theta(z_t \mid z_{t+1}, \boldsymbol{x}_{1:t})}$. That is, the current state $z_t$ depends on future observations $\boldsymbol{x}_{t+1:T}$ only through the next state $z_{t+1}$.

  - ⬛ Step 1a We show $\boxed{p_\theta(z_t \mid z_{t+1}, \boldsymbol{x}_{1:t+1}) = p_\theta(z_t \mid z_{t+1}, \boldsymbol{x}_{1:t})}$.

    $$
    \begin{aligned}
    p_\theta(z_t \mid z_{t+1}, \boldsymbol{x}_{1:t+1}) &= p_\theta(z_t \mid z_{t+1}, \boldsymbol{x}_{t+1}, \boldsymbol{x}_{1:t}) && \text{split off term from sequence} \\
    &= \frac{p_\theta(z_t, \boldsymbol{x}_{t+1} \mid z_{t+1}, \boldsymbol{x}_{1:t})}{p_\theta(\boldsymbol{x}_{t+1} \mid z_{t+1}, \boldsymbol{x}_{1:t})} && \text{conditional density} \\
    &= \frac{p_\theta(\boldsymbol{x}_{t+1} \mid \cancel{z_t, z_{t+1}}, \boldsymbol{x}_{1:t})\, p(z_t \mid z_{t+1}, \boldsymbol{x}_{1:t})}{p_\theta(\cancel{\boldsymbol{x}_{t+1}} \mid z_{t+1}, \boldsymbol{x}_{1:t})} && \text{chain rule} \\
    &= p(z_t \mid z_{t+1}, \boldsymbol{x}_{1:t}) && \text{FPOBN}
    \end{aligned}
    $$

    In the last line, the two canceled terms are equal by FPOBN (the Fundamental Property of Bayes Networks).[3]

  - ⬛ Step 1b We show $\boxed{p_\theta(z_t \mid z_{t+1}, \boldsymbol{x}_{1:t+2}) = p_\theta(z_t \mid z_{t+1}, \boldsymbol{x}_{1:t+1})}$. By the same argument as in step 1a (splitting up the sequence, conditional density, chain rule), but replacing

    $$\boldsymbol{x}_{1:t+1} \leftarrow \boldsymbol{x}_{1:t+2} \quad , \quad \boldsymbol{x}_{t+1} \leftarrow \boldsymbol{x}_{t+2}$$

    the proposition holds if

    $$p(\boldsymbol{x}_{t+2} \mid z_t, z_{t+1}, \boldsymbol{x}_{1:t+1}) = p(\boldsymbol{x}_{t+2} \mid z_{t+1}, \boldsymbol{x}_{1:t+1})$$

    that is if we get the same cancelation. And we see

    $$
    \begin{aligned}
    p(\boldsymbol{x}_{t+2} \mid z_t, z_{t+1}, \boldsymbol{x}_{1:t+1}) &= \sum_{k=1}^K p(\boldsymbol{x}_{t+2}, z_{t+2} = k \mid z_t, z_{t+1}, \boldsymbol{x}_{1:t+1}) && \text{LTP} \\
    &= \sum_{k=1}^K p(\boldsymbol{x}_{t+2} \mid z_{t+2} = k, \cancel{z_t}, z_{t+1}, \boldsymbol{x}_{1:t+1})\, p(z_{t+2} = k \mid \cancel{z_t}, z_{t+1}, \boldsymbol{x}_{1:t+1}) && \text{chain rule, FPOBN} \\
    &= \sum_{k=1}^K p(\boldsymbol{x}_{t+2}, z_{t+2} = k \mid z_{t+1}, \boldsymbol{x}_{1:t+1}) && \text{undo chain rule} \\
    &= p(\boldsymbol{x}_{t+2} \mid z_{t+1}, \boldsymbol{x}_{1:t+1}) && \text{undo LTP}
    \end{aligned}
    $$

---

[2]Our proof is inspired by the proof given by Hamilton (1994, pp.700-702) for the `ARHMM` (i.e, the special case of `rAR-HMM` in which there are no recurrent edges).

[3]The Fundamental Property of Bayes Networks is: *A node is independent of its non-descendants given its parents.* In particular, since $z_t$ is a non-descendent of $\boldsymbol{x}_{t+1}$, it is independent of $\boldsymbol{x}_{t+1}$ given its parents $z_{t+1}$ and $\boldsymbol{x}_{1:t}$.

– $\boxed{\text{Conclusion}}$ The claim follows from Steps 1a and 1b by an induction argument.

- $\boxed{\text{Step 2.}}$ We show that $\boxed{\underbrace{p(z_t, z_{t+1} \mid \boldsymbol{x}_{1:T})}_{\text{smoothed pairwise}} = \underbrace{p(z_{t+1} \mid \boldsymbol{x}_{1:T})}_{\text{smoothed}} \underbrace{p(z_{t+1} \mid z_t, \boldsymbol{x}_{1:t})}_{\text{transition}} \frac{\overbrace{p(z_t \mid \boldsymbol{x}_{1:t})}^{\text{filtered}}}{\underbrace{p(z_{t+1} \mid \boldsymbol{x}_{1:t})}_{\text{predicted}}}}$ . We have

$$\begin{aligned} p(z_t, z_{t+1} \mid \boldsymbol{x}_{1:T}) &= p(z_{t+1} \mid \boldsymbol{x}_{1:T})\, p(z_t \mid z_{t+1}, \boldsymbol{x}_{1:T}) && \text{chain rule} \\ &= p(z_{t+1} \mid \boldsymbol{x}_{1:T})\, p(z_t \mid z_{t+1}, \boldsymbol{x}_{1:t}) && \text{Step 1} \\ &= p(z_{t+1} \mid \boldsymbol{x}_{1:T})\, \frac{p(z_t \mid \boldsymbol{x}_{1:t})\, p(z_{t+1} \mid z_t, \boldsymbol{x}_{1:t})}{p(z_{t+1} \mid \boldsymbol{x}_{1:t})} && \text{Bayes rule (on 2nd term)}^4 \end{aligned}$$

- $\boxed{\text{Step 3.}}$ We prove the proposition.

$$\begin{aligned} p(z_t \mid \boldsymbol{x}_{1:T}) &= \sum_{k=1}^{K} p(z_t, z_{t+1} = k \mid \boldsymbol{x}_{1:T}) && \text{Law of Total Prob.} \\[2mm] \boldsymbol{\xi}_{t \mid T} &= \sum_{k=1}^{K} \Big[\boldsymbol{\xi}_{t+1 \mid T}\Big]_k \Big[\boldsymbol{A}_t\Big]_{:,z_{t+1}=k} \frac{\boldsymbol{\xi}_{t \mid t}}{\Big[\boldsymbol{\xi}_{t+1 \mid t}\Big]_k} && \text{Step 2, Notation} \\[2mm] &= \boldsymbol{\xi}_{t \mid t} \sum_{k=1}^{K} \Big[\boldsymbol{A}_t\Big]_{:,z_{t+1}=k} \frac{\Big[\boldsymbol{\xi}_{t+1 \mid T}\Big]_k}{\Big[\boldsymbol{\xi}_{t+1 \mid t}\Big]_k} && \text{Pull out constant} \\[2mm] &= \boldsymbol{\xi}_{t \mid t} \odot \left\{ \boldsymbol{A}_t^T \cdot \Big[\boldsymbol{\xi}_{t+1 \mid T} \,(\div)\, \boldsymbol{\xi}_{t+1 \mid t}\Big] \right\} && \text{Def. matrix multiplication} \end{aligned}$$

$\square$

**Remark A.2.2.** As we saw in Step 1, the derivation of the smoother in Prop. A.2.2 relies on the fact that while each node (observation or state) can have *many* observation parents, it can have only *one* state parent (namely, the closest in time from the present or past). $\triangle$

**Remark A.2.3.** The filtering (Prop A.2.1) and smoothing (Prop A.2.2) formulae reveal that state estimation for `rAR-HMM` can be handled for :

- *any order* of recurrence and/or autoregression[5]

- *any functional form* of emissions and transitions

Furthermore, although it was not explicitly represented here, the same formulae hold when there are

- Modulation of transitions and emissions by *exogenous covariates.*[6]

$\triangle$

# B HRSDM Method Details

We now review modeling and inference details for our proposed `HSRDM`, in the following sections

---

[4]To justify the application of Bayes rule, imagine that $z_t$ plays the role of the parameter and $z_{t+1}$ plays the role of the observed data. The term $\boldsymbol{x}_{1:t}$ is just a conditioning set throughout.

[5]In fact, the proof reveals that the order can increase with timestep $t$, opening the door to constructions involving exponential weighted moving averages.

[6]A sequence of vectors $\{\boldsymbol{u}_t\}$ is considered to be a sequence of exogenous covariates if each $\boldsymbol{u}_t$ contains no information about $z_t$ that is not contained in $\boldsymbol{x}_{1:t-1}$ (Hamilton, 1994, pp.692).

## B.1 Priors on model parameters

The symbol $\theta$ denotes all model parameters for our HSRDM. Using the structure of our model in Equation 2.1, we can expand $\theta$ into constituent components: $\theta = (\theta_{\texttt{ss}}, \theta_{\texttt{es}}, \theta_{\texttt{ee}}, \theta_{\texttt{init}})$, where $\theta_{\texttt{ss}}$ are the parameters that govern the system-level discrete state transitions, $\theta_{\texttt{es}}$ govern the entity-level discrete state transitions, $\theta_{\texttt{ee}}$ govern the entity-level emissions, and $\theta_{\texttt{init}}$ govern the initial distribution for states and regimes.

We define a prior over $\theta$ whose factorization structure reflects this decomposition:

$$p(\theta) = p(\theta_{\texttt{ss}})\, p(\theta_{\texttt{es}})\, p(\theta_{\texttt{ee}})\, p(\theta_{\texttt{init}}) \tag{B.1}$$

As we see in Sec. B.4, this choice of prior simplifies the M-step.

**Prior on system-level state transition parameters $\theta_{\texttt{ss}}$.** For the system-level transition probability matrix $\mathbf{\Pi}$, a $L \times L$ matrix whose entries are all non-negative and rows sum to one, we assume a sticky Dirichlet prior (Fox et al., 2011) to encourage self-transitions so that in typical samples, one system state would persist for long segments. Concretely, for each row we set

$$\mathbf{\Pi}_{j1}, \ldots \mathbf{\Pi}_{jL} \sim \text{Dir}(\alpha, \ldots \alpha, \alpha + \kappa, \alpha, \ldots \alpha) \tag{B.2}$$

where all $L$ entries have a symmetric base value $\alpha = 1.0$, and the added value $\kappa$ that impacts the self-transition entry (the $(j, j)$-th entry of the matrix) is set to 10.0. We then set the log transition probability $\widetilde{\mathbf{\Pi}}$ to the element-wise log of $\mathbf{\Pi}$.

**Prior on entity-level state transition parameters $\theta_{\texttt{es}}$.** In our experiments, we used a non-informative prior, $p(\theta_{\texttt{es}}) \propto 1$. The use of a sticky Dirichlet prior, as was used with the system-level transition parameters, could be expected to produce smoother entity-level state segmentations. Currently, the entity-level segmentations are choppier than those at the system-level (e.g., compare the bottom-left and top-right subplots of Fig. 5).

**Prior on emissions $\theta_{\texttt{ee}}$.** In our experiments, we used a non-informative prior, $p(\theta_{\texttt{ee}}) \propto 1$.

**Prior on initial states and observations $\theta_{\texttt{init}}$.** For initial states at both system and entity level, we use a symmetric Dirichlet with large concentration so that all states have reasonable probability a-priori. This avoids the pathology of ML estimation that locks into only one state as a possible initial state early in inference due to poor initialization.

## B.2 Updating the posterior over system-level states

In this section, we discuss the update to the posterior over system-level states; that is, the variational E-S step of Equation 3.2. We find

$$q(s_{0:T}) \widetilde{\propto} \exp \left\{ \underbrace{\log \pi_s(s_0)}_{\text{init dist}} + \underbrace{\sum_{t=1}^{T} \log p(s_t \mid s_{t-1}, \boldsymbol{x}_{t-1}^{(1:J)}, \theta)}_{\text{transitions}} + \underbrace{\sum_{j=1}^{J} \sum_{t=1}^{T} \mathbb{E}_{q(z_{0:T}^{(1:J)})} \log p(z_t^{(j)} \mid z_{t-1}^{(j)}, \boldsymbol{x}_{t-1}^{(j)}, s_t, \theta)}_{\text{emissions}} \right\}$$

$$= \underbrace{\pi_s(s_0)}_{\text{init state}} \underbrace{\prod_{t=1}^{T} p(s_t \mid s_{t-1}, \boldsymbol{x}_{t-1}^{(1:J)}, \theta)}_{\text{transitions}} \underbrace{\prod_{j=1}^{J} \exp \left\{ \sum_{k=1}^{K} \log \pi_{z_j}(z_0^{(j)} = k)\, q(z_0^{(j)} = k) \right\}}_{\text{initial emissions}}$$

$$\underbrace{\prod_{j=1}^{J} \prod_{t=1}^{T} \exp \left\{ \sum_{k,k'=1}^{K} \log p(z_t^{(j)} = k' \mid z_{t-1}^{(j)} = k, \boldsymbol{x}_{t-1}^{(j)}, s_t, \theta)\, q(z_t^{(j)} = k', z_{t-1}^{(j)} = k) \right\}}_{\text{remaining emissions}} \tag{B.3}$$

This can be considered as the posterior of an input-output Hidden Markov Model with $J$ independent autoregressive categorical emissions. The evaluation of the transition function is $\mathcal{O}\big(T(L^2 + LDJM_s)\big)$, where $M_s$ is the dimension of the system-level covariates, and where we have assumed that the evaluation of the system-level recurrence function $g$ takes $DJM_s$ operations, as it would if $g_\psi\big(\boldsymbol{x}_{t-1}^{(1:J)}, \boldsymbol{v}_{t-1}\big) = (\boldsymbol{x}_{t-1}^{(1)}, \ldots, \boldsymbol{x}_{t-1}^{(J)}, \boldsymbol{v}_{t-1})^T$. The evaluation of the emissions function is $\mathcal{O}\big(TJL(K^2 + KDM_e)\big)$, where $M_e$ is the dimension of the entity-level covariates, and where we have assumed that the evaluation of the entity-level recurrence function $f$ takes $DM_e$ operations, as it would if $f_\phi\big(\boldsymbol{x}_{t-1}^{(j)}, \boldsymbol{u}_{t-1}^{(j)}\big) = (\boldsymbol{x}_{t-1}^{(j)}, \boldsymbol{u}_{t-1}^{(j)})^T$. Thus, by Props. A.2.1 and A.2.2, filtering

and smoothing can be computed with $\mathcal{O}\big(TJ\big[L^2 + L(K^2 + DM_s + KDM_e)\big]\big)$ runtime complexity, under mild assumptions on the recurrence functions. As a result, so can the computation of the unary and adjacent pairwise marginals necessary for the VES and M steps.

## B.3 Updating the posterior over entity-level states

In this section, we discuss the update to the posterior over entity-level states; that is, the variational E-Z step of Equation 3.2. We obtain

$$q(z_{0:T}^{(1:J)}) \widetilde{\propto} \prod_{j=1}^{J} \Big[ \underbrace{\pi_{z_j}(z_0^{(j)})}_{\text{initial state } (\in \mathbb{R}^K)} \underbrace{\prod_{t=1}^{T} \exp\Big\{ \sum_{\ell=1}^{L} \log p(z_t^{(j)} \mid z_{t-1}^{(j)}, \boldsymbol{x}_{t-1}^{(j)}, s_t = \ell, \theta)\, q(s_t = \ell) \Big\}}_{\text{transitions } (\in \mathbb{R}^{(\widetilde{T}-1) \times K \times K})}$$

$$\underbrace{p(\boldsymbol{x}_0^{(j)} \mid z_0^{(j)}, \theta)}_{\text{initial emission } (\in \mathbb{R}^K)} \underbrace{\prod_{t=1}^{T} p(\boldsymbol{x}_t^{(j)} \mid \boldsymbol{x}_{t-1}^{(j)}, z_t^{(j)}, \theta)}_{\text{remaining emissions } (\in \mathbb{R}^{(\widetilde{T}-1) \times K})} \Big] \tag{B.4}$$

where we have defined $\widetilde{T} \triangleq T + 1$ to denote all timesteps after accounting for the zero-indexing. This variational factor can be considered as posterior of $J$ conditionally independent Hidden Markov Models with autoregressive categorical emissions which recurrently feedback into the transitions. As per Sec. B.2, the transition function can be evaluated with $\mathcal{O}\big(TJL(K^2 + KDM_e)\big)$ runtime complexity, where $M_e$ is the dimension of the entity-level covariates, and where we have assumed that the evaluation of the entity-level recurrence function $f$ takes $DM_e$ operations, as it would if $f_\phi\big(\boldsymbol{x}_{t-1}^{(j)}, \boldsymbol{u}_{t-1}^{(j)}\big) = (\boldsymbol{x}_{t-1}^{(j)}, \boldsymbol{u}_{t-1}^{(j)})^T$. The evaluation of the emissions function is $\mathcal{O}\big(TJKD^2\big)$, assuming that the emissions distribution has a density that can be evaluated with $\mathcal{O}\big(D^2\big)$ operations at each timestep. Thus, by Props. A.2.1 and A.2.2, filtering and smoothing can be computed with $\mathcal{O}\big(TJ\big[K^2 + KD^2 + KDM_e\big]\big)$ runtime complexity, under mild assumptions on the entity-level recurrence function and the emissions distribution. As a result, so can the computation of the unary and adjacent pairwise marginals necessary for the VEZ and M steps.

## B.4 Updating the parameters

The M-step updates the transition parameters and emission parameters $\theta$ of our HSRDM given recent estimates of state-level posterior $q(s_{0:T})$ and entity-level posteriors $q(z_{0:T}^{(1:J)})$.

This update requires solving the following optimization problem

$$\theta = \arg\max_{\theta} \mathcal{L}(\theta)$$

$$\text{where } \mathcal{L}(\theta) \triangleq \underbrace{\mathbb{E}_{q(s_{0:T}^{(1:J)})q(z_{0:T}^{(1:J)})}\Big[ \log p(\boldsymbol{x}_{0:T}^{(1:J)}, s_{0:T}^{(1:J)}, z_{0:T}^{(1:J)} \mid \theta) \Big]}_{\text{expected log complete data likelihood}} + \underbrace{\log p(\theta)}_{\text{log prior}} \tag{B.5}$$

Based on the structure of the model in Equation 2.1, we can decompose this into separate optimization problems over the different model pieces $\theta = (\theta_{\text{ss}}, \theta_{\text{es}}, \theta_{\text{ee}}, \theta_{\text{init}})$ by assuming an appropriately factorized prior, as was done in Equation B.1.

To be concrete, for a HSRDM with transitions given by Equation 2.2 and Gaussian vector autoregressive (Gaussian VAR) emissions

$$\boldsymbol{x}_t^{(j)} \mid \boldsymbol{x}_{t-1}^{(j)}, z_t^{(j)} \sim N\Big( \boldsymbol{A}_j^{(z_t^{(j)})} \boldsymbol{x}_{t-1}^{(j)} + \boldsymbol{b}_j^{(z_t^{(j)})},\ \boldsymbol{Q}_j^{(z_t^{(j)})} \Big), \tag{B.6}$$

as used in Secs. 5.1 and 5.2 we have

$$\theta_{\text{ss}} = (\boldsymbol{\Lambda}, \widetilde{\boldsymbol{\Pi}}), \quad \theta_{\text{es}} = \{\boldsymbol{\Psi}_j, \widetilde{\boldsymbol{P}}_j\}_{j=1}^{J}, \quad \theta_{\text{ee}} = \{\{\boldsymbol{A}_{jk}, \boldsymbol{b}_{jk}, \boldsymbol{Q}_{jk}\}_{k=1}^{K}\}_{j=1}^{J}$$

Using this grouping of the parameters along with the complete data likelihood specification of Equation 2.1 and the prior assumption in Equation B.1, we can decompose the objective as

$$\mathcal{L}(\theta) = \mathcal{L}_{\text{init}}(\theta_{\text{init}}) + \mathcal{L}_{\text{ss}}(\theta_{\text{ss}}) + \sum_{j=1}^{J} \mathcal{L}_{\text{es}}^{(j)}(\theta_{\text{es}}^{(j)}) + + \mathcal{L}_{\text{ee}}^{(j)}(\theta_{\text{ee}}^{(j)})$$

We can then complete the optimization by separately performing M-steps for each of the subcomponents of $\theta$. For example, to optimize the parameters governing the entity-level discrete state transitions $\theta_{\mathsf{es}}^{(j)}$ for each entity $j = 1, \ldots, J$, we only need to optimize

$$
\mathcal{L}_{\mathsf{es}}^{(j)}(\theta_{\mathsf{es}}^{(j)}) \triangleq \underbrace{\sum_{t=1}^{T} \mathbb{E}_{q(z_{0:T}^{(1:J)})q(s_{0:T})} \left[ \log p(z_t^{(j)} \mid z_{t-1}^{(j)}, \boldsymbol{x}_{t-1}^{(j)}, s_t, \theta_{\mathsf{es}}) \right]}_{\text{expected log entity discrete state transitions}} + \underbrace{\log p(\theta_{\mathsf{es}})}_{\text{log prior}}
$$

$$
= \sum_{t=1}^{T} \sum_{k,k'=1}^{K} \sum_{\ell=1}^{L} \log p(z_t^{(j)} = k' \mid z_{t-1}^{(j)} = k, \boldsymbol{x}_{t-1}^{(j)}, s_t = \ell, \theta_{\mathsf{es}}) \, q(z_t^{(j)} = k', z_{t-1}^{(j)} = k) \, q(s_t = \ell)
$$

$$
+ \log p(\theta_{\mathsf{es}}) \tag{B.7}
$$

In particular, we do not require the variational posterior over the full entity-level discrete state sequence $q(z_{0:T}^{(1:J)})$, but merely the pairwise marginals $q(z_t^{(j)} = k', z_{t-1}^{(j)} = k)$, obtainable from the VEZ step in Equation 3.2. Similarly, we do not require the variational posterior over the full system-level discrete state sequence $q(s_{0:T})$, but merely the unary marginals $q(s_t = \ell)$, obtainable from the VES step in Equation 3.2.

The other components of $\theta$ are optimized similarly. In general, the optimization can be performed by gradient descent (e.g. using JAX for automatic differentiation (Bradbury et al., 2018)), although it can be useful to bring in closed-form solutions for the M substeps in certain special cases. For instance, when Gaussian VAR emissions are used as in Equation B.6, the entity emission parameters $\theta_{\mathsf{ee}} = \{\{\boldsymbol{A}_{jk}, \boldsymbol{b}_{jk}, \boldsymbol{Q}_{jk}\}_{k=1}^{K}\}_{j=1}^{J}$ can be estimated with closed-form updates using the sample weights $q(z_t^{(j)} = k)$ available from the VEZ-step.

## B.5 Variational lower bound

A lower bound on the marginal log likelihood $\texttt{VLBO} \leq \log p(\boldsymbol{x}_{0:T}^{(1:J)} \mid \theta)$, is given by

$$
\texttt{VLBO}\,[\theta, q] = \underbrace{\mathbb{E}_q \left[ \log p(\boldsymbol{x}_{0:T}^{(1:J)}, z_{0:T}^{(1:J)}, s_{0:T} \mid \theta) \right]}_{\text{energy}} + \underbrace{\mathbb{H}\left[ q(z_{0:T}^{(1:J)}, s_{0:T}) \right]}_{\text{entropy}} \tag{B.8}
$$

The energy term $\mathbb{E}_q \left[ \log p(\boldsymbol{x}_{0:T}^{(1:J)}, z_{0:T}^{(1:J)}, s_{0:T} \mid \theta) \right]$ is identical to the relevant term in the objective function for the M-step given in Equation B.5. Based on the structure of the model assumed in Equation 2.1, the energy term decomposes into separate pieces for initialization, system transitions, entity transitions, and emissions. For example, see Equation B.7 for the piece relevant to entity transitions.

Now we consider computation of the entropy $\mathbb{H}\left[ q(z_{0:T}^{(1:J)}, s_{0:T}) \right]$ in Equation B.8. Since the variational factors $q(s_{0:T})$ and $\{q(z_{0:T}^{(j)})\}_{j=1}^{J}$ given respectively by the VES step in Sec. B.2 and the VEZ step in Sec. B.3 both have the form of $\texttt{rARHMMs}$, we can compute the entropy $\mathbb{H}\left[ q(z_{0:T}^{(1:J)}, s_{0:T}) \right] = \sum_{j=1}^{J} \mathbb{H}\left[ q(z_{0:T}^{(j)}) \right] + \mathbb{H}\left[ q(s_{0:T}) \right]$, using for each individual term the entropy for HMMs provided in Eq. 12 of Hughes et al. (2015).

## B.6 Smart initialization

We can construct a "smart" (or data-informed) initialization of a $\texttt{HSRDM}$ via the following two-stage procedure:

1. We fit $J$ *bottom-level* $\texttt{rARHMMs}$, one for each of the $J$ entities. In particular, the emissions for each bottom-level $\texttt{rAR-HMM}$ are the emissions of the full $\texttt{HSRDM}$ given in Equation 2.3, and the transitions are the entity-level transitions given in Equation 2.2b.

2. We fit one *top-level* $\texttt{ARHMM}$. Here, the $J$ emissions are the entity-level transitions given in Equation 2.2b. The transitions are the system-level transitions given in Equation 2.2a. The observations are taken to be the most-likely entity-level states as inferred by the bottom-level $\texttt{rARHMMs}$.

Below we give details on these initializations. In particular, both the bottom-level and top-level models *themselves* need initializations. We use the term *pre-initialization* to refer to the initializations of those models.

### B.6.1 Initialization of bottom-level `rARHMMs`

Here we fit $J$ *bottom-level* `rARHMMs`, one for each of the $J$ entities, independently. In particular, the emissions for each bottom-level `rAR-HMM` are the emissions of the full `HSRDM` given in Equation 2.3, and the transitions are the entity-level transitions given in Equation 2.2b.

**Pre-initialization.** The $J$ bottom-level `rARHMMs` *themselves* need good (data-informed) initializations. As an example, we describe the pre-initialization procedure in the particular case of Gaussian VAR emissions, as given in Equation B.6. In particular, we focus on a strategy for pre-initializing these emission parameters $\{\boldsymbol{A}_j^{(k)}, \boldsymbol{b}_j^{(k)}, \boldsymbol{Q}_j^{(k)}\}_{j,k}$, since the higher-level parameters in the model can be learned via the two-stage initialization procedure.

In particular, for each $j = 1, \ldots, J$,

    a) We assign the observations $\boldsymbol{x}_{1:T}^{(j)}$ to one of $K$ states by applying the $K$-means algorithm to either the observations themselves or to their velocities (discrete derivatives) $\boldsymbol{x}_{2:T}^{(j)} - \boldsymbol{x}_{1:T-1}^{(j)}$, depending upon user specification. We use the former choice in the FigureEight data, and the latter choice for basketball data.

    b) We then initialize the parameters by running separate vector autoregressions within each of the $K$ clusters. In particular, for each state $k = 1, \ldots, K$,

        a) We find state-specific observation matrix $\boldsymbol{A}_j^{(k)}$ and biases $\boldsymbol{b}_j^{(k)}$ by applying a (multi-outcome) linear regression to predict $\boldsymbol{x}_t^{(j)}$ from the $\boldsymbol{x}_{t-1}^{(j)}$ whenever $\boldsymbol{x}_t^{(j)}$ belongs to the $k$-th cluster.

        b) We estimate the regime-specific covariance matrices $\boldsymbol{Q}_j^{(k)}$ from the residuals of the above vector autoregresssion.

We initialize the entity-level transition parameters $\{\boldsymbol{\Psi}_j, \widetilde{\boldsymbol{P}}_j\}_{j=1}^J$ to represent a sticky transition probability matrix. This implies that we initialize $\boldsymbol{\Psi}_j = \boldsymbol{0}$ for all $j$.

**Expectation-Maximization.** After pre-initialization, we estimate the $J$ independent `rARHMMs` by using the expectation maximization algorithm. Posterior state inference (i.e. the E-step) for this procedure is justified in Sec. A.2. Note that the posterior state inference for these bottom-level `rARHMMs` can be obtained by reusing the VEZ step of Equation B.4 by setting the number of system states to $L = 1$.

### B.6.2 Initialization of top-level `ARHMM`

Here we fit a *top-level* `ARHMM`. In particular, the emissions for the `ARHMM` are the entity-level transitions of the `HSRDM` given in Equation 2.2b, and the transitions of the `ARHMM` are the system-level transitions given in Equation 2.2a. We can perform posterior state inference for the top-level `ARHMM` by reusing the VES step of Equation B.3 with inputs being the posterior state beliefs on $z_{0:T}^{(1:J)}$ from the bottom-level `rARHMMs`.

## B.7 Multiple Examples

In some datasets, we may observe the same $J$ entities over several distinct intervals of synchronous interaction. We call each separate interval of contiguous interaction an "example". For example, the raw basketball dataset from Sec. 5.2 is organized as a collection of separate plays, where each play is one separate example. Between the end of one play and the beginning of the next, the players might have changed positions entirely, perhaps even having gone to the locker room and back for halftime.

Let $E$ be the number of examples. Each example, indexed by $e \in \{1, 2, \ldots E\}$, starts at some reference time $\tau_e$ and has $T_e$ total timesteps, covering the time sequence $t \in \{\tau_e, \tau_e + 1, \ldots, \tau_e + T_e\}$. We'll model each per-example observation sequence $\boldsymbol{x}_{\tau_e:\tau_e+T_e}^{(1:J)}$ as an iid observation from our `HSRDM` model.

To efficiently represent such data, we can stack the observed sequences for each example on top of one another. This yields a total observation sequence $\boldsymbol{x}_{0:T}^{(1:J)}$ that covers all timesteps across all examples, defining

$T = T_1 + T_2 + \ldots + T_E$. This representation doesn't waste any storage on unobserved intervals between examples, easily accommodates examples of arbitrarily different lengths, and integrates well with modern vectorized array libraries in our Python implementation. As before in the single example case, our computational representation of $\boldsymbol{x}_{0:T}^{(1:J)}$ is as a 3-d array with dimensionality $(T, J, D)$.

For properly handling this compact representation, bookkeeping is needed to track where one example sequence ends and another begins. We thus track the ending indices of each example in this stacked representation: $\mathcal{E} = \{t_0, t_1, \ldots, t_{E-1}, t_E\}$, where $-1 = t_0 < t_1 < t_2 < \ldots < t_{E-1} < t_E = T$, and where $t_e = \tau_e + T_e$ is the last valid timestep observed in the $e$-th example for $e = 1, \ldots, E$.

By inspecting the inference updates gives above, including the filtering and smoothing updates for `rAR-HMM` (see Props. A.2.1 and A.2.2), we find that we can handle this situation as follows:

- *E-steps (VEZ or VES)*: Whenever we get to a cross-example boundary, we replace the usual transition function with an initial state distribution. More concretely, the transition function for the VES step in Equation B.3 is modified so that any timestep $t$ that represents the start of a new example sequence (that is, satisfies $t - 1 \in \mathcal{E}$) is replaced with $\pi_s$, and the transition function for the VEZ step in Equation B.4 at such timesteps is replaced with $\pi_{z_j}$. Similarly, the emissions functions at such timesteps are replaced with the initial emissions. This maneuver can be justified by noting that for any timestep $t$ designating the onset of a new example, the initial state distributions play the role of $\boldsymbol{A}_t$ and the initial emissions play the role of $\boldsymbol{\epsilon}_t$ in Props. A.2.1 and A.2.2.

- *M-steps*: Due to the model structure, the objective function $\mathcal{L}$ for the M-step can be expressed as a sum over timestep-specific quanities; for example, see Equation B.7. Thus, in the case of multiple examples, we simply adjust the set of timesteps over which we sum in the objective functions relative to each M substep. We update the entity emissions parameters $\theta_{\texttt{ee}}$ by altering the objective to sum over timesteps that *aren't* at the beginning of an example (so we sum over timesteps $t$ where $t - 1 \notin \mathcal{E}$). We update the system state parameters $\theta_{\texttt{ss}}$ and entity state parameters $\theta_{\texttt{es}}$ by altering the objectives to sum only over timesteps that haven't straddled an example transition boundary. That is, we want to ignore any pair of timesteps $(t, t+1)$ where $t \in \mathcal{E}$, so we again sum only over timesteps $t$ where $t - 1 \notin \mathcal{E}$. Finally, we update the initialization parameters $\theta_{\texttt{init}}$ by altering the objective to sum over all timesteps that *are* at the beginning of an example.

## C    Forecasting Methodology Details

Here we detail how we assess model fit (Sec. C.1) and compute forecasts (Sec. C.2). The primary difference between fitting and forecasting is that only the former has access to observations from evaluated entities over a time interval of interest. Hence, a good fit is more easily attained. A good forecast requires predictions of the discrete latent state dynamics without access to future observations, whereas fitting can use the future observations to infer the discrete latent state dynamics. However, model fit is still useful to investigate; for instance, it can be useful to determine if piecewise linear dynamics (including the choice of $K$, the number of per-entity states) provide a good model for a given dataset.

### C.1    Model fit

To compute the fit of the model to $\{\boldsymbol{x}_t^{(j)}, \ldots \boldsymbol{x}_{t+u}^{(j)}\}$, the $j$-th entity's observed time series over some slice of integer-valued timepoints $[t, \ldots, t+u]$, we initialize

$$\boldsymbol{\mu}_{t-1}^{(j)} = \boldsymbol{x}_{t-1}^{(j)}$$

And then forward simulate. In particular, for time $\tau$ in $[t, \ldots, t+u]$, we do

$$\boldsymbol{\mu}_\tau^{(j)} \triangleq \sum_{k=1}^{K} q(z_\tau^{(j)} = k) \boldsymbol{\mu}_{\tau,k}^{(j)} \tag{C.1}$$

where $\boldsymbol{\mu}_{\tau,k}^{(j)}$ is the conditional expectation of the emissions distribution from Equation 2.3 with Radon-Nikodỳm density $p(\boldsymbol{x}_\tau^{(j)} \mid \boldsymbol{x}_{\tau-1}^{(j)} = \boldsymbol{\mu}_{\tau-1}^{(j)}, z_\tau^{(j)})$. For example, with Gaussian vector autoregressive (VAR) emissions, we have

$$\boldsymbol{\mu}_{\tau,k}^{(j)} \triangleq \boldsymbol{A}_{j,k} \, \boldsymbol{\mu}_{\tau-1,k}^{(j)} + \boldsymbol{b}_{j,k}$$

The resulting sequence $\{\boldsymbol{\mu}_t^{(j)}, \ldots \boldsymbol{\mu}_{t+u}^{(j)}\}$ gives the variational posterior mean for the $j$-th entity's observed time series over timepoints $[t, \ldots, t+u]$.

## C.2 Partial forecasting

By partial forecasting, we mean predicting $\{\boldsymbol{x}_t^{(j)}, \ldots \boldsymbol{x}_{t+u}^{(j)}\}_{j \in \mathcal{J}}$, the observed time series from some **to-be-forecasted entities** (with indices $\mathcal{J} \subset [1, \ldots, J]$) over some forecasting horizon of integer-valued timepoints $[t, \ldots, t+u]$, given observations $\{\boldsymbol{x}_t^{(j)}, \ldots \boldsymbol{x}_{t+u}^{(j)}\}_{j \in \mathcal{J}^c}$ from the **contextual entities** $\mathcal{J}^c \triangleq \{j \in [1, \ldots, J] : j \notin \mathcal{J}\}$ over that same forecasting horizon, as well as observations from all entities over earlier time slices $\{\boldsymbol{x}_0^{(j)}, \ldots \boldsymbol{x}_{t-1}^{(j)}\}_{j \in [1, \ldots, J]}$.

To instantiate partial forecasting, we must first adjust inference, and then perform a forward simulation.

1. *Inference adjustment.* The VEZ step (Sec. B.3) is adjusted so that the variational factors on the entity-level states over the forecasting horizon $\{q(z_t^{(j)}, \ldots, z_{t+u}^{(j)})\}_j$ are computed only for the contextual entities $\{j \in \mathcal{J}^c\}$. Likewise, the VES step (Sec. B.2) is adjusted so that the variational factor on the system-level states over the forecasting horizon $q(s_t, \ldots, s_{t+u})$ is computed from the observations $\{\boldsymbol{x}_t^{(j)}, \ldots \boldsymbol{x}_{t+u}^{(j)}\}_j$ and estimated entity-level states $\{q(z_t^{(j)}, \ldots, z_{t+u}^{(j)})\}_j$ only from the contextual entities $\{j \in \mathcal{J}^c\}$. As a result, the M-step on the system-level parameters $\theta_{\text{ss}}$ automatically exclude information from the to-be-forecasted entities $\mathcal{J}$ over the forecasting horizon $[t, \ldots, t+u]$.

2. *Forward simulation.* Using the adjusted inference procedure from Step 1, we can use the Viterbi algorithm (or some other procedure) to obtain estimated system-states $\{\hat{s}_t, \ldots, \hat{s}_{t+u}\}$ that do not depend on information from the to-be-forecasted entities $\mathcal{J}$ over the forecasting horizon $[t, \ldots, t+u]$. We then make forecasts by forward simulating. In particular, for time $\tau$ in $[t, \ldots, t+u]$, we sample

$$z_t^{(j)} \sim p(z_t^{(j)} \mid z_{t-1}^{(j)}, \boldsymbol{x}_{t-1}^{(j)}, \hat{s}_t, \theta) \tag{C.2}$$

$$\boldsymbol{x}_t^{(j)} \sim p(\boldsymbol{x}_t^{(j)} \mid \boldsymbol{x}_{t-1}^{(j)}, z_t^{(j)}, \theta) \tag{C.3}$$

for all to-be-forecasted entities $j \in \mathcal{J}$.

Note in particular that the dependence of Equation C.2 upon $\hat{s}_t$ allows our predictions about to-be-forecasted entities $\{j \in \mathcal{J}\}$ to depend upon observations from the contextual entities $\{j \in \mathcal{J}^c\}$ over the forecasting horizon.

# D FigureEight: Experiment Details, Settings, and Results

## D.1 Data generating process

**Example D.1.1.** (*FigureEight.*) Consider a model where we directly observe continuous observations $\boldsymbol{x}_{0:T}^{(1:J)}$, and where each $\boldsymbol{x}_t^{(j)} \in \mathbb{R}^2$ lives in the plane (i.e. $D = 2$). We form "Figure Eights" by having the observed dynamics rotate around an "upper circle" $\mathcal{C}_1$ with unit radius and center $\breve{\boldsymbol{b}}^{(1)} \triangleq (0,1)^T$ and a "lower circle" $\mathcal{C}_2$ with unit radius and center $\breve{\boldsymbol{b}}^{(2)} \triangleq (0,-1)^T$. Entities tend to persistently rotate around one of these circles; however, when the observation approaches the intersection of the two circles $\mathcal{C}_1 \cap \mathcal{C}_2 = \{(0,0)\}$, recurrent feedback can shift the entity's dynamics into a new state (the other circle). These shifts occur only when the system-level state has changed; these shifts are not predictable from the entity-level time series alone. In

particular, we have

$$\underbrace{s_t \mid s_{t-1}}_{\text{system transitions}} = h(s_t) \tag{D.1a}$$

$$\underbrace{z_t^{(j)} \mid z_{t-1}^{(j)}, \boldsymbol{x}_{t-1}^{(j)}, s_t}_{\text{entity transitions}} \sim \text{Cat-GLM}_K\left( \boldsymbol{\eta}_t^{(j)} = \underbrace{\boldsymbol{\Psi}^{(s_t)} f(\boldsymbol{x}_{t-1}^{(j)})}_{\text{recurrence}} + \underbrace{\widetilde{\boldsymbol{P}}_j^T \boldsymbol{e}_{z_{t-1}^{(j)}}}_{\text{transitions}} \right) \tag{D.1b}$$

$$\underbrace{\boldsymbol{x}_t^{(j)} \mid \boldsymbol{x}_{t-1}^{(j)}, z_t^{(j)}}_{\text{observation dynamics}} \sim N\left( \boldsymbol{A}_j^{(z_t^{(j)})} \boldsymbol{x}_{t-1}^{(j)} + \boldsymbol{b}_j^{(z_t^{(j)})}, \boldsymbol{Q}_j^{(z_t^{(j)})} \right) \tag{D.1c}$$

Here, the notation used follows that of Equation 2.2. Each line of this true data-generating process is explained in the corresponding paragraph below.

**System-level state transitions.** We take the number of system states to be $L = 2$. We set the system state chain $\{s_t\}_{t=1}^T$ through a deterministic process $h$ which alternates states every 100 timesteps. We emphasize that in the *true* data-generating process, there is no recurrent feedback from observations $x$ to system states $s$.

**Entity-level state transitions.** We set entity-specific baseline transition preferences to be highly sticky, $\boldsymbol{P}_j = \begin{bmatrix} p & (1-p) \\ (1-p) & p \end{bmatrix}$, where $p$ is close to 1.0 (concretely, $p = .999$). By design, these preferences can be overridden when an entity travels near the origin. We choose the recurrence transformation $f : \mathbb{R}^D \to \mathbb{R}$ to be the radial basis function $f(\boldsymbol{x}) = \kappa \exp(-\frac{\|\boldsymbol{x}\|_2^2}{2\sigma^2})$, which returns a large value when the observation $\boldsymbol{x}_{t-1}^{(j)}$ is close to the origin. Similarly, we set the weight vector for these recurrent features to nudge observations near the origin to the system-preferred state. We set $\boldsymbol{\Psi}^{(\ell)} \in \mathbb{R}^K$ so entry $(\boldsymbol{\Psi}^{(\ell)})_k = a_{\text{high}}$ if the entity-level state $k$ is preferred by the system-level state $\ell$, and $a_{\text{low}}$ otherwise, with $a_{\text{high}} \gg a_{\text{low}}$. Concretely, We set $a_{\text{high}} = 2$ and $a_{\text{low}} = -2$.

**Emissions.** To construct the entity-level emission distributions for each state (indexed by $k$), we choose $\boldsymbol{A}_j^{(k)} = \boldsymbol{A}_j$ to be a rotation matrix with angle $\theta = (-1)^r \frac{2\pi}{\tau_j}$ for all entity-level states $k$, where $\tau_j$ is the entity-specific periodicity and $r \in \{0, 1\}$ determines the rotation direction. We may use a rotation matrix $\boldsymbol{A}$ to rotate the observation around a center $\breve{\boldsymbol{b}}$, by constructing dynamics of the form $\boldsymbol{A}(\boldsymbol{x} - \breve{\boldsymbol{b}}) + \breve{\boldsymbol{b}}$; therefore, to construct circle centers that are specific to entity-level states using Equation D.1c, we set $\boldsymbol{b}_j^{(k)} = (\boldsymbol{I} - \boldsymbol{A}_j) \breve{\boldsymbol{b}}^{(k)}$ for all entities $j$ and all entity-level states $k$. We set each of the observation noise covariance matrices $\boldsymbol{Q}_j^{(k)}$ to be diagonal, with diagonal entries equal to 0.0001.

We simulate data from the *FigureEight* model (Example D.1.1), where there are $J = 3$ entities, each with $T = 400$ observations, where the periodicities for each entity are given by $(\tau_1, \tau_2, \tau_3) = (5, 20, 40)$. $\triangle$

## D.2 Models and Hyperparameter Tuning

**HSRDM.** We fit our `HSRDM` with transitions given in Equation 2.2 and Gaussian vector autoregressive emissions as in Equation B.6. We set $L = K = 2$. We set the entity-level recurrence $f$ to a Gaussian radial basis function and no system-level recurrence $g$. We use a sticky Dirichlet prior on system-level transition parameters ($\alpha = 1$ and $\kappa = 10$). For 'smart' initialization, the bottom-half of the model is trained for 5 iterations while the top-half is trained for 20 iterations. For the K-Means algorithm, the random state is set to seed 120. The model is then trained for 10 CAVI iterations with 50 iterations per M-Step. We don't tune any specific hyperparameters of our HSRDM. The model is run across 5 independent initializations (initialization seeds 120-124). Five forecasting samples are generated per trial (sample seeds 120-124). The optimal MSE across all initialization trials and forecasting samples is recorded. The average MSE for the trial containing the best sample is taken across all samples to demonstrate diversity in sample generation.

**rAR-HMM.** A collection of $J$ `rAR-HMM` models can be fit as a special case of a `HSRDM` model where the number of system states is taken to be $L = 1$. We train three rAR-HMM models with separate strategies: ("*Indep.*", "*Pool*", and "*Concat.*"). For 'smart' initialization, the bottom-half of the models are trained for 5 iterations while the top-halves are trained for 20 iterations. For the K-Means algorithm, the random state is

set to seed 120. The models are then trained for 10 CAVI iterations with 50 iterations per M-Step. We don't tune any specific hyperparameters of the rAR-HMM. The models are run across 5 independent initializations (seeds 120-124). Five forecasting samples are generated per trial (sample seeds 120-124). The optimal MSE across all initialization trials and forecasting samples is recorded for each model. The average MSE for the trial containing the best sample is taken across all samples from that trial to demonstrate diversity in sample generation.

**DSARF.**   We train the Deep Switching Autoregressive Factorization model ([Farnoosh et al., 2021](#)) with several different parameters. We train with several different choices of lags ($l$) and spatial factors ($K$), where the number of discrete states ($S = 2$) is fixed to match our data-generation. For each strategy ("*Indep.*", "*Pool*", and "*Concat.*"), we conduct a grid search across combinations of $l$ and $K$, specifically for $l = \{1, ..., i\}, \forall i \in \{1, ..., 10\}$ and $K \in \{1, ..., 10\}$. The hyperparameters that produce the optimal MSE value are selected for further experimentation. DSARF *Indep.* is optimal with spacial factors $K$ and lags $L$: $K = 3, L = \{1, 2\}$; DSARF *Pool* is optimal with $K = 9, L = \{1, 2, 3, 4, 5, 6, 7\}$; and DSARF *Concat.* is optimal with $K = 5, L = \{1, 2\}$.

Once hyperparameters are selected, each model (and strategy) is trained with 500 epochs and a learning rate of 0.01, across 10 independent random seeds (seeds 120-129) representing different initializations and forecasts. Two seeds produce clear outlier results (one in *Indep.* and one in *Concat.*) from very poor initializations and were excluded from the average results in Fig. 2, as they would have drastically shifted the mean (in a direction that demonstrates worse perofrmance for the model). Post training for each model, we use the learned parameters to draw long-term forecasts.

### D.3   Recurrence Ablation

In Fig. 2, the rAR-HMM baselines are system state ablations for the `HSRDM`. Here, we include a recurrence ablation for the *FigureEight* task. The best MSE across 5 independent random initializations is 0.05 and the Ave. MSE with standard error for the random initialization across forecasting samples is 2.41(0.78). The Mdn. MSE across the independent trial sample averages is 2.41. `HSRDM` without recurrence still performs worse than the `HSRDM` with recurrence. See the trajectory below in Fig. D.1.

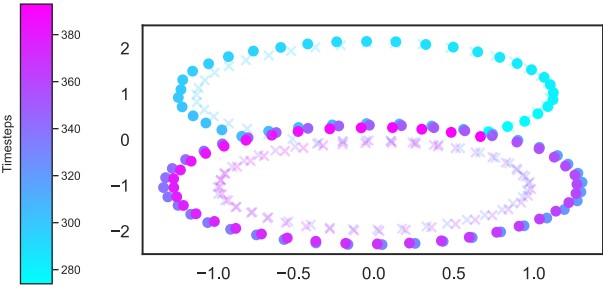

Figure D.1: *FigureEight* No Recurrent Feedback

## E   Basketball: Experiment Details, Settings, and Results

### E.1   Dataset

**Raw dataset.**   We obtain NBA basketball player location data for 636 games within the 2015-2016 NBA season from a publicly available repo ([Linou, 2016](#)). Each sample provides the quarter of the game, number of seconds left in quarter, time on shot clock, (x,y,z) location of ball, and the (x,y) locations and IDs for the 10 players on the court. The court is represented as the rectangle $[0, 96] \times [0, 50]$ in the space of squared feet.

**Selection of games.**   We focus on modeling the dynamics in games involving the Cleveland Cavaliers (CLE), the 2015-2016 NBA champions. In particular, out of 40 available games containing CLE, we investigate the 31

games containing one of the four most common starting lineups: 1. `K. Irving - L. James - K. Love - J. Smith - T. Thompson`; 2. `K. Irving - L. James - K. Love - T. Mozgov - J. Smith`; 3. `L. James - K. Love - T. Mozgov - J. Smith - M. Williams`; 4. `M. Dellavedova - L. James - K. Love - T. Mozgov - J. Smith`. Two games had data errors (lack of tracking or event data), which left a total of $G = 29$ games for analysis.

**Downsampling.** The raw data is sampled at 25 Hz. Following Alcorn & Nguyen (2021), we downsample to 5 Hz.

**From plays to examples.** The raw basketball dataset is represented in terms of separate *plays* (e.g. `shot block, rebound offense, shot made`). Following Alcorn & Nguyen (2021), we preprocess the dataset so that these plays are non-overlapping in duration. We also remove plays that do not contain one of CLE's four most common starting lineups. For the purpose of unsupervised time series modeling, we then convert the plays into coarser-grained observational units. Although plays are useful for the classification task pursued by Alcorn & Nguyen (2021), play boundaries needn't correspond to abrupt transitions in player locations. For example, the player coordinates are essentially continuous throughout `shot block -> rebound offense -> shot made` sequence mentioned above. Hence, we concatenate consecutive plays from the raw dataset until there is an abrupt break in player motion and/or a sampling interval longer than the nominal sampling rate. These observational units are called *events* in the main body of the paper (Sec. 5.2). Functionally, these observational units serve as *examples* (Sec. B.7). That is, when training models, each example is treated as an i.i.d. sample from the assumed model. For the remainder of the Appendix, we refer to these observational units as examples.

By construction, examples have a longer timescale than the plays in the original dataset. Examples typically last between 20 seconds and 3 minutes. For comparison, a `rebound offense` play takes a fraction of a second.

At the implementational level, we infer an example boundary whenever at least one condition below is met in a sequence of observations:

1. The wall clock difference between timesteps is larger than 1.2 times the nominal sampling rate.

2. The player's step size on the court (given by the discrete derivative between two timesteps) is abnormally large with respect to either the court's length or width, where abnormally large is defined as having an absolute z-score larger than 4.0.

**Court rotation.** The location of a team's own basket changes at half time. This can switch can alter the dynamics on the court. We would like to control for the direction of movement towards the offensive and defensive baskets, as well as for player handedness. To control for this, we assume that the focal team (CLE)'s scoring basket is always on the left side of the court. When it is not, we rotate the court 180 degrees around the center of the basketball court. (Equivalently, we negate both the x and y coordinates with respect to the center of the court.) Since the basketball court has a width of 94 feet and a length of 50 feet, its center is located at $(47, 25)$ when orienting the width horizontally. We prefer this normalization strategy to the random rotations strategy of Alcorn & Nguyen (2021), because the normalization strategy allows us to learn different dynamics for offense (movement to the left) and defense (movement to the right).

**Index assignments.** Each sample from our dataset gives the coordinates on the court of 10 players. Here we describe how we map the players to entity indices. Recall that we only model the plays that consist of starters from a focal team, CLE. We assign indices 0-4 to represent CLE starters, and indices 5-9 to represent opponents.

Index assignment for CLE is relatively straightforward. Although we model plays from the $G$ games involving four different starting lineups, we can consistently interpret the indices as `0: Lebron James, 1: Kevin Love, 2: J.R. Smith, 3: Starting Center, 4: Starting Guard`. Depending on the game, the starting center was either T. Mazgov or T. Thompson. Similarly, the starting guard was either K. Irving, M. Williams, or M. Dellavedova.

Index assignment for the opponents is more involved. The opponent teams can vary from game to game, and even a fixed team substitutes players throughout a game. There are numerous mechanisms for assigning indices in the face of such *player substitutions* (Raabe et al., 2023). Although role-based representations are popular (e.g. see (Felsen et al., 2018) or (Zhan et al., 2019)) because they capture invariants lost within identity-based representations (Lucey et al., 2013), we used a simple heuristic whereby we assign indices 5-9 based on the the player's typical positions. The typical positions can be scraped from Wikipedia. We let the model discover dynamically shifting roles for the players via its hierarchical discrete state representation.

One complication in assigning indices from these position labels is that the provided labels commonly blend together multiple positions (e.g. 'Shooting guard / small forward' or 'Center / power forward'). Should the second player be labeled as a center or a forward? What if there are multiple centers? How do we discriminate between two forwards? To solve such problems, we proceed as follows, operating on a play-by-play basis

1. *Assign players to coarse position groups.* We first assign players to coarse position groups (forward, guard, center). We assume that each play has 2 forwards, 1 center, and 2 guards. We use indices 5-6 to represent the forwards, index 7 to represent the center, and indices 8-9 to represent the guards. As noted above, a given player can be multiply classified into a coarse position group; however, a reasonable assignment for a player can be made by considering the position labels for the other players who are on the court at the same time. To do this, we form $B$, a $5 \times 3$ binary matrix whose rows are players on the team and whose columns represent the coarse position groups. An entry in the matrix is set to True if the player is classified into that position group. We start with the rarest position group (i.e. the column in $B$ with the smallest column sum) and assign players to that position group, starting with players who have the least classifications (i.e. the players whose rows in $B$ have the smallest row sum). Ties are broken randomly. We continue until we have satisfied the specified assignments (2 forwards, 1 center, and 2 guards). If it is not possible to make such coarse assignments, we discard the play from the dataset.

2. *Order players within coarse position groups.* This step is only needed for forwards and guards; there is only 1 ordering of the single center. We use an arbitrary ordering of forward positions:

   ```
   FORWARD_POSITIONS_ORDERED = [
       "Small forward / shooting guard",
       "Small forward / point guard",
       "Small forward",
       "Small forward / power forward",
       "Power forward / small forward",
       "Power forward",
       "Power forward / center",
       "Center / power forward",
       "Shooting guard / small forward",
   ]
   ```

   and guard positions by

   ```
   GUARD_POSITIONS_ORDERED = [
       "Small forward / shooting guard",
       "Shooting guard / small forward",
       "Shooting guard",
       "Shooting guard / point guard",
       "Point guard / shooting guard",
       "Point guard",
       "Combo guard",
   ]
   ```

   For each players assigned to a position group in $\{forward, guard\}$, we order the players in terms of their location of their position on the above lists. Ties are broken randomly.

**Normalization**   To assist with initialization and learning of parameters, we normalize the player locations on the court from the rectangle $[0, 96] \times [0, 50]$ in units of feet to the unit square $[0, 1] \times [0, 1]$.

## E.2 Models and Hyperparameter Tuning

**HSRDM.** Here we model $J = 10$ basketball player trajectories on the court with an HSRDM with Gaussian vector autoregressive emissions; that is, we use

$$\underbrace{s_t \mid s_{t-1}, \boldsymbol{x}_{t-1}^{(1:J)}}_{\text{system transitions}} \sim \text{Cat-GLM}_L \Big( \underbrace{\widetilde{\boldsymbol{\Pi}}^T \boldsymbol{e}_{s_{t-1}}}_{\text{endogenous transition preferences}} + \underbrace{\boldsymbol{\Lambda} \, g_\psi \big( \boldsymbol{x}_{t-1}^{(1:J)}, \boldsymbol{v}_{t-1} \big)}_{\text{bias from recurrence and covariates}} \Big) \tag{E.1}$$

$$\underbrace{z_t^{(j)} \mid z_{t-1}^{(j)}, \boldsymbol{x}_{t-1}^{(j)}, s_t}_{\text{entity transitions}} \sim \text{Cat-GLM}_K \Big( \underbrace{(\widetilde{\boldsymbol{P}}_j^{(s_t)})^T \boldsymbol{e}_{z_{t-1}^{(j)}}}_{\text{endogenous transition preferences}} + \underbrace{\boldsymbol{\Psi}_j^{(s_t)} f_\phi \big( \boldsymbol{x}_{t-1}^{(j)}, \boldsymbol{u}_{t-1}^{(j)} \big)}_{\text{bias from recurrence and covariates}} \Big) \tag{E.2}$$

$$\underbrace{\boldsymbol{x}_t^{(j)} \mid \boldsymbol{x}_{t-1}^{(j)}, z_t^{(j)}}_{\text{observation dynamics}} \sim N \Big( \boldsymbol{A}_j^{(z_t^{(j)})} \boldsymbol{x}_{t-1}^{(j)} + \boldsymbol{b}_j^{(z_t^{(j)})}, \, \boldsymbol{Q}_j^{(z_t^{(j)})} \Big) \tag{E.3}$$

where $\boldsymbol{x}_t^{(j)} \in \big([0,1] \times [0,1]\big)$ gives player $j$'s location on the normalized basketball court at timestep $t$.

Our system-level recurrence $g_\psi\big(\boldsymbol{x}_{t-1}^{(1:J)}, \boldsymbol{v}_{t-1}\big) = \boldsymbol{x}_{t-1}^{(1:J)}$ reports *all* player locations $\boldsymbol{x}_{t-1}^{(1:J)}$ to the system-level transition function, allowing the probability of latent game states to depend on player locations. Inspired by Linderman et al. (2017), our entity-level recurrence function $f_\phi\big(\boldsymbol{x}_{t-1}^{(j)}, \boldsymbol{u}_{t-1}^{(j)}\big) = (\boldsymbol{x}_{t-1}^{(j)}, \mathbb{I}[\boldsymbol{x}_{t-1,0}^{(j)} < 0.0], \mathbb{I}[\boldsymbol{x}_{t-1,0}^{(j)} > 1.0], \mathbb{I}[\boldsymbol{x}_{t-1,1}^{(j)} < 0.0], \mathbb{I}[\boldsymbol{x}_{t-1,1}^{(j)} > 1.0])^T$, where $\boldsymbol{x}_{t,d}^{(j)}$ is the $d$-th coordinate of $\boldsymbol{x}_t^{(j)}$ and $\mathbb{I}[\cdot]$ is the indicator function, reports an individual player's location $\boldsymbol{x}_{t-1}^{(j)}$ (and out-of-bounds indicators) to that player's entity-level transition function, allowing each player's probability of remaining in autoregressive regimes to vary in likelihood over the court.

We set the number of system and entity states to be $L = 5$ and $K = 10$ based on informal experimentation with the training set; we leave formal setting of these values based on the validation set to future work. For the sticky Dirichlet prior on system-level transitions, as given in Equation B.2, we set $\alpha = 1.0$ and $\kappa = 50.0$ so that the prior would put most of its probability mass on self-transition probabilities between .90 and .99.

We initialize the model using the smart initialization strategy of Sec. B.6. We pre-initialize the entity emissions parameters $\theta_{\text{ee}}$ by applying the $k$-means algorithm to each player's discrete derivatives (so long as consecutive timesteps do not span an example boundary). We pre-initialize the entity state parameters $\theta_{\text{es}}$ by setting $\widetilde{\boldsymbol{P}}$ to be the log of a sticky symmetric transition probability matrix with a self-transition probability of 0.90, and by drawing the entries of $\boldsymbol{\Psi}$ i.i.d from a standard normal. We pre-initialize the system state parameters $\theta_{\text{ss}}$ by setting $\widetilde{\boldsymbol{\Pi}}$ to be the log of a sticky symmetric transition probability matrix with a self-transition probability of 0.95, and by drawing the entries of $\boldsymbol{\Lambda}$ i.i.d from a standard normal. We pre-initialize the initialization parameters $\theta_{\text{init}}$ by taking the initial distribution to be uniform over system states, uniform over entity states for each entity, and standard normal over initial observations for each entity and each entity state. We execute the two-stage initialization process via 5 iterations of expectation-maximization for the $J$ bottom-half rARHMMs, followed by 20 iterations for the top-half ARHMM.

We run our CAVI algorithm for 2 iterations, as informal experimentation with the training set suggested this was sufficient for approximate ELBO stabilization.

**rARHMMs.** By ablating the top-level discrete "game" states (i.e., the system-level switches) in the HSRDM, we obtain independent rARHMMs (Linderman et al., 2017), one for each of the $J = 10$ players. More specifically, by removing the system transitions in Equation E.1 from the model, the entity transitions simplify as $p(z_t^{(j)} \mid z_{t-1}^{(j)}, \boldsymbol{x}_{t-1}^{(j)}, s_t) = p(z_t^{(j)} \mid z_{t-1}^{(j)}, \boldsymbol{x}_{t-1}^{(j)})$, because the entity transition parameters simplify as $\widetilde{\boldsymbol{P}}_j^{(s_t)} = \widetilde{\boldsymbol{P}}_j$ and $\boldsymbol{\Psi}_j^{(s_t)} = \boldsymbol{\Psi}_j$. As a result, the $J$ bottom-level rARHMMs are decoupled. Implementationally, this procedure is equivalent to an HSRDM with $L = 1$ system states. Initialization and training is otherwise performed identically as with HSRDM.

**HSDM.** By ablating the multi-level recurrence from the `HSRDM`, we obtain a *hierarchical switching dynamical model* (`HSDM`). This can be accomplished by setting $g_\psi \equiv 0$ in Equation E.1 and $f_\phi \equiv 0$ in Equation E.2. Initialization and training is otherwise performed identically as with `HSRDM`.

**Agentformer.** AgentFormer (Yuan et al., 2021) is a multi-agent (i.e. multi-entity) variant of a transformer model whose forecasts depend upon both temporal and social (i.e. across-entity) relationships. Unless otherwise noted, we follow Yuan et al. (2021) in determining the training hyperparameters. In particular, our prediction model consists of 2 stacks of identical layers for the encoder and decoder with a dropout rate of 0.1. The dimensions of keys, queries and timestamps for the agentformer are set to 16, while the hidden dimension of the feedforward layer is set to 32. The number of heads for the multi-head agent aware attention is 8 and all MLPs in the model have a hidden dimension of (512,256). The latent code dimension of the CVAE is set to 32, and the agent connectivity threshold is set to 100. Because the basketball training datasets have many more examples than the pedestrian trajectory prediction experiments in Yuan et al. (2021) (which only have 8 examples), we train the agentformer model and the DLow trajectory sampler for 20 epochs each (rather than 100) to keep the computational load manageable. We therefore apply the Adam optimizer with learning rate of $10^{-3}$ rather than $10^{-4}$ to accommodate the reduced number of epochs. Also, to match the specifications of the evaluation strategy from Sec. E.3, we set the number of future prediction frames during training to 30, and the number of diverse trajectories sampled by the trajectory sampler to 20. We ensure convergence by tracking the mean-squared error.

**SNLDS.** At suggestion of an anonymous reviewer, we also compared to a recent non-linear model which does not coordinate interactions across entities, but provides discrete and continuous latent variables as well recurrent feedback: `SNLDS` (Dong et al., 2020).[7] `SNLDS` inherently models only entity-level time series by design; it is not originally designed to model many interacting entities. We tried the Independence entity-to-system strategy (see Sec. 5.1 of our paper) to adapt it to model many entities. The precise hyperparameter settings can be found by inspecting the `basketball.yaml` configurations file used to run `SNLDS` (see our codebase for link). Unless otherwise noted, we follow Ansari et al. (2021)'s modeling of the electricity duration dataset to determine the training hyperparameters (e.g. trainable covariance matrices, a transformer within the emissions network with the same network size, etc.). Paralleling the electricity duration dataset, we set the dimensionality of the latent continuous state (4) to be double that of the observed variables (2). For consistency with the `HSRDM`, we set the number of entity states to be $K = 10$. We also allowed multi-level recurrence to both the continuous and discrete latent chains to parallel the multi-level recurrence of the `HSRDM`. We trained each basketball player's model for 20,000 iterations (using the first 1,000 as warmup iterations for setting the learning rate). We ensure convergence by examining trace plots to verify that the ELBO-based training objective converged after many epochs, and by verifying that reconstructions of past data looked reasonable.

### E.3 Evaluation strategy

We divide the $G = 29$ total games into 20 games to form a candidate training set, 4 games to form a validation set (for setting hyperparameters), and 5 games to form a test set. Of the first 20 games within our candidate training set, we construct small (1 game), medium (5 games), and large (20 games) training sets. The small, medium, and large training sets contained 20, 215, and 676 examples, respectively.

The test set contained 158 examples overall. However, we required that each example be at least 10 seconds long (i.e. 50 timesteps) to be included in the evaluation run. This exclusion criterion left $E = 75$ examples. For each such example, we uniformly select a timepoint $T^* \in [T_{\text{min-context-length}}, T - T_{\text{forecast-length}}]$ to demarcate where the context window ends. We set $T_{\text{min-context-length}} = 4$ seconds (i.e. 20 timesteps) and $T_{\text{forecast-length}} = 6$ seconds (i.e. 30 timesteps). The first $[0, T^*]$ seconds are shown to the trained model as context, and forecasts are made within the forecasting window of $\mathcal{F} := [T^* + 1, T^* + T_{\text{forecast-length}}]$ seconds.

---

[7]We used the open source implementation of the `SNLDS` model provided by a successor package for RED-SDS (Ansari et al., 2021): `https://github.com/abdulfatir/REDSDS`. We chose this because (1) this is the newest suggested codebase, so it should be the most "state-of-the-art", and (2) as the newest codebase, it should be easier to install and maintain. (Older code from 2017 is often tough to get working on newer hardware, such as the non-Intel chips of newer Macbooks.)

For a fixed example $e$, forecasting sample $s$, player $j$, and forecasting method $m$, we summarize the error in a forecasted trajectory by **mean forecasting error** (MFE)

$$\text{MFE}_{m;e,s,j} \triangleq \frac{1}{|\mathcal{F}|} \sum_{t \in \mathcal{F}} \sqrt{\sum_{d=0}^{1} (\widehat{x}_{e,t,j,d,m,s} - x_{e,t,j,d})^2} \tag{E.4}$$

where $x_{e,t,j,d}$ is the true observation on example $e$ at time $t$ for player $j$ on court dimension $d$, and $\widehat{x}_{e,t,j,d,m,s}$ is the forecasted observation by forecasting sample $s$ using forecasting method $m$. So $\text{MFE}_{m;e,s,j}$ gives the average distance over the forecasting window between the forecasted trajectory and the true trajectory.

To quantify performance of forecasting methods, we define a model's *example-wise* mean forecasting error as

$$\text{MFE}_{m;e} \triangleq \frac{1}{SJ} \sum_{s=1}^{S} \sum_{j=1}^{J} \text{MFE}_{m;e,s,j} \tag{E.5}$$

Taking the mean of $\text{MFE}_{m;e}$ and its standard error lets us quantify a model's typical squared forecasting error on an example, as well as our uncertainty, with

$$\text{MFE}_m \triangleq \frac{1}{E} \sum_{e=1}^{E} \text{MFE}_{m;e}, \qquad \sigma(\text{MFE}_m) \triangleq \frac{\sqrt{\sum_{e=1}^{E} (\text{MFE}_{m;e} - \text{MFE}_m)^2}}{E} \tag{E.6}$$

Although in Sec. E.1, we described normalization of basketball coordinates to the unit square for the purpose of model initialization and training, when evaluating models, we convert the forecasts and ground truth back to unnormalized coordinates, so that MFE has units of feet. That is, we represent observations $x_{e,t,j,d}$ and forecasts $\widehat{x}_{e,t,j,d,m,s}$ on the basketball court (of size $[0, 94] \times [0, 50]$ feet). Thus $\sqrt{\text{MSE}_m}$ can be interpreted as a model's typical amount of error in feet on the court at a typical timepoint in the forecasting window (but of course forecasting error tends to be lower at timepoints closer to $T^*$ than farther from $T^*$).

# F  Marching Band: Experiment Details, Settings, and Results

## F.1  Data generating process

We introduce *MarchingBand*, a synthetic dataset consisting of individual marching band players ("entities") moving across a 2D field in a coordinated routine to visually form a sequence of letters. Each observation is a position $\boldsymbol{x}_t^{(j)} \in \mathbb{R}^2$, with the unit square centered at $(0.5, 0.5)$ representing the field. Each entity's position over time on the unit square follows the current letter's prescribed movement pattern perturbed by small-scale iid zero-mean Gaussian noise. Each state is stable for 200 timesteps before transitioning in order to the next state. When reaching a field boundary, typically the player is reflected back in bounds. However, with some small chance, an entity will continue out-of-bounds (OOB, $\boldsymbol{x}_{t1} \notin (0, 1)$). When enough players become OOB (up to a user-controlled threshold), this bottom-up signal triggers the current system state immediately to a special "come together and reset" state, denoted "C". During the next 50 timesteps, all players move to the center, then return to repeat the most recent letter before continuing on to remaining letters. Note that the true data-generating process does not come from an HSRDM generative model.

For our experiments, we set $J = 64$, the OOB threshold to be 11, and the letter sequence to spell "LAUGH". We use $N = 10$ independent examples for fitting, where each sequence is of a different length according to the number of "C" states triggered. We observe corresponding time lengths for each sequence: $\{1000, 1050, 1050, 1000, 1000, 1000, 1050, 1050, 1000, 1100\}$. The total time $T$ across all sequences is 10300, where end-times for each example are provided to the model as $\{1000, 2050, 3100, 4100, 5100, 6100, 7150, 8200, 9200, 10300\}$.

## F.2  Models and Hyperparameter Tuning

**HSRDM (with and without recurrence).**  For all methods, we fairly provide knowledge of the true number of system states, $S = 6 = \{L, A, U, G, H, C\}$. Models with system states (HSRDM and its recurrent ablation) set $S = 6$ and have $K = 4$ entity states. We set system-level recurrence $g$ to count the number

of entities out of bounds and the entity-level recurrence $f$ to the identity function. The emission model is set to a Gaussian Autoregressive. All system and entity states are initially set to uniform distributions, and we use a sticky Dirichlet prior on system-level transition parameters ($\alpha = 1$ and $\kappa = 10$). For 'smart' initialization, the bottom-half and top-half of the model are only trained for a single iteration, such that little to no initialization is used for this experiment. The K-Means algorithm is set to a random state with a seed 120. The model is then trained for 10 CAVI iterations with 50 iterations per M-Step. We don't tune any specific hyperparameters. The model is run across 5 independent initializations (with recurrence: seeds 120-123, 126 and without recurrence: seeds 120-124). Seeds (such as 124 for the HSRDM with recurrence) that do not lead to *any* effective optimization (NAN values) are discarded and an alternative seed is attempted. This procedure is the same for all competitor models, which also tend to produce ineffective training due to randomly poor initializations. The best system-state classification accuracy across all trials is recorded for the models.

**rAR-HMM.** A collection of $J$ `rAR-HMM` models can be fit as a special case of a `HSRDM` model where the number of system states is taken to be $L = 1$. The number of entity states is taken to be $K = 6$ with a uniform prior over possible states. For 'smart' initialization, the bottom-half and top-half of the model are only trained for a single iteration, such that little to no initialization is used for this experiment. The K-Means algorithm is set to a random state with a seed 120. The model is then trained for 10 CAVI iterations with 50 iterations per M-Step. We don't tune any specific hyperparameters of the rAR-HMM. The model is trained across 5 independent initializations (seeds 120-122 and 124-126). The resulting most likely states for each entity at each timepoint are clustered with a K-Means algorithm (random state seed = 120), and are compared with the ground truth system-level accuracy for each initialization. The best classification accuracy is recorded.

**DSARF.** We train the Deep Switching Autoregressive Factorization model (Farnoosh et al., 2021) with several different parameters. We train with several different choices of lags ($l$) and spatial factors ($K$), where the number of discrete states ($S = 6$) is fixed to match our data-generation. We use the '*Concat.*') strategy, and conduct a grid search across combinations of $l$ and $K$, specifically for $l = \{1, ..., i\}, \forall i \in \{1, ..., 10, 200\}$ and $K \in \{1, ..., 10, 15, 25, 30\}$. These numbers for search were selected based on prior work in Ref. (Farnoosh et al., 2021), such that our values match the hyperparameters used for datasets with a similar size. Also, the selections are based on the knowledge that our dataset changes discrete states around every 200 time-steps. The hyperparameters that produce the optimal system classification accuracy are selected for further experimentation, which we observe to be: $l = l = \{1, ..., 200\}$ and $K = 25$.

Once hyperparameters are selected, each model is trained with 200 epochs (as indicated to be sufficient in the hyperparameter search) and a learning rate of 0.01, across 10 independent random initializations (seeds 120-129). The resulting most likely states for each entity at each timepoint are clustered with a K-Means algorithm (random state seed = 120), and are compared with the ground truth system-level accuracy for each initialization. The best classification accuracy is recorded.

## F.3 Experiments with up to 200 entities

To demonstrate that the `HSRDM` can train and learn the system-level hidden states in the *MarchingBand* dataset with a larger number of entities $J > 64$, we run an experiment with $J = 200$ entities. To simplify our experiment, we left out the cluster state C that gets triggered when a certain number of entities go out of bounds. Instead, we only have $S = 5 = \{L, A, U, G, H\}$. All other variables (number of dimensions ($D = 2$), priors, initialization procedure, and training procedure) are kept the same as in the $J = 64$ case. Across 5 initializations, we obtain a best system state classification accuracy of 100%, a median accuracy of 100%, and a average accuracy of 90%. Two out of the five initializations only obtained an accuracy of about $\approx 70\%$, which brought down the total average. Varying results based on unlucky initializations for this type of model is common.

### F.4 Experiments on Larger Feature Dimensions

Experiments in the main paper focus on interpretable modeling of data with $D = 2$ feature dimensions recorded each timestep. To demonstrate that the HSRDM can train and learn the system-level hidden states in the *MarchingBand* dataset with a larger number of dimensions $D > 2$, we run experiments with $D = \{10, 30\}$ dimensions. While $D = 2$ is the true $(x, y)$ position of the marching band player, $D = \{10, 30\}$ are the true positions with added gaussian noise $\mathcal{N}(0, 0.0004)$. To simplify this experiment, we left out the cluster state C that gets triggered when a certain number of entities go out of bounds. Instead, we only have $S = 5 = \{L, A, U, G, H\}$. All other variables (number of entities ($J = 64$), priors, initialization procedure, and training procedure) are kept the same as in the $J = 64$ case.

Across 5 initializations, the average and standard deviation system level classification accuracies for $D = \{2, 10, 30\}$ are shown in Fig. F.1. We observe that the best classification accuracies remain high for all $D$; however, the average decays as $D$ increases ($D = 2 : 88\%, D = 10 : 80\%, D = 30 : 75\%$). Since all $D$ sizes were run with the same number of initialization iterations, number of training examples, and the same number of CAVI iterations, we hypothesize that $D = 30$ might achieve better performance with more careful tuning of these hyper-parameters. In Fig. F.1, we also show the run times for a single trial of each $D$ size in minutes for an entire training iteration.

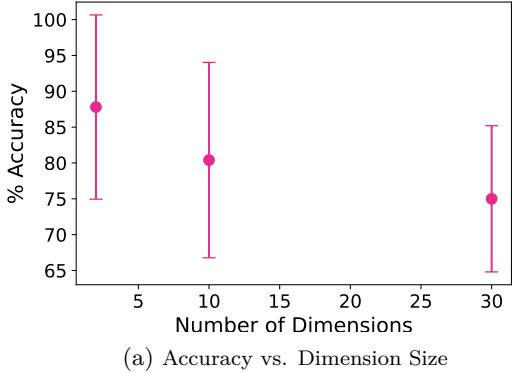 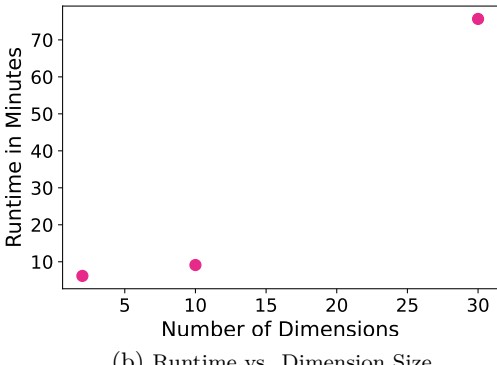

(a) Accuracy vs. Dimension Size     (b) Runtime vs. Dimension Size

Figure F.1: *MarchingBand* system level accuracy and runtimes for $D = \{2, 10, 30\}$.

## G    Soldiers: Experiment Details, Settings, and Results

For the visual security experiment, based on a quick exploratory analysis, we set $K = 4$ and $L = 3$. For the sticky Dirichlet prior on system-level transitions, as given in Equation B.2, we set $\alpha = 1.0$ and $\kappa = 50.0$, so that the prior would put most of its probability mass on self-transition probabilities between .90 and .99.

