# OpenReview forum: "Discovering group dynamics in coordinated time series via hierarchical recurrent switching-state models"
_TMLR — Accepted by TMLR_

### Review · Reviewer_Pthk · 2025-03-12

**Summary Of Contributions:**

This paper proposes Hierarchical Switching Recurrent Dynamical Model (HSRDM), a state-space model with only discrete latent variables, which is parameter efficient and is aimed to improve interpretability of the learned dynamics. The model assumes independent autoregressive hidden markov models, where an additional discrete latent variable induces changes in the global dynamics. Experiments on both synthetic and real-world data demonstrate the improved interpretability of global and individual system dynamics, and show decent forecasting results despite model simplicity.

**Audience:**

Yes

**Claims And Evidence:**

Yes

**Requested Changes:**

**Text**

- The "optional exogenous covariates" mentioned in Eqs. 2.2ab $u_{t-1}^j$ and $v_{t-1}$ might be a bit confusing. Would it be possible to add one sentence connecting these to the example of including the ball's position in NBA?
- In section 2, the following statement is imprecise:
> If we remove the top-level system states ... our HSRMD reduces to Linderman's recurrent autoregressive HMM.

  I believe the statement is not correct, as Linderman does not consider individual element. You should specify this per-entity modelling in your case to avoid confusions. For example, you can talk about some per-entity rAR-HMM instead.

**Experiments**

For acceptance, I consider the following point should be addressed. From my understanding, it might be expected to see worse results, but still, it is critical to understand whether HSRDM suffers in terms of forecasting performance despite improved interpretability and parameter efficiently.

- Only 3 baselines are present: Agentformer and rAR-HMM in NBA; and rAR-HMM and DSARF in MarchingBrand. It would be very important to include additional baselines to understand **whether the improved interpretability comes at the cost of forecasting performance.** Below are some baselines that could be used. All of them have available code.
  - NRI from Kipf et al. (2018) [1]: It would be interesting to check how does your work compare to interective systems, considering this work is mentioned in your related work section.
  - At least 1 baseline that considers Switching Dynamical Systems should be present. Considering you already have rAR-HMM, it might be interesting to consider rSLDS from Linderman et al. (2017) [2], or one of SNLDS (Dong et al. 2020) [3], RED-SDS (Ansari et al. 2021) [4].
  - There is this interesting recent paper, Graph Swicthing Dynamical Systems (GRASS) (Liu et al 2023 [5]), which considers SDSs with individual elements, similar to your case. Here a comparison might be very interesting to highlight the advantages of your method in terms of interpretability.

Below are some miscellaneous points:

- Would it be possible to try the NBA experiment replacing Eq. (2.3) with a Recurrent Neural Network instead?
-  All the experiments consider 2D trajectories. Would it be possible to devise a synthetic experiment with increasing $D$ to understand the problems outlined in **Section 6: Limitations**?

**References**

[1] Kipf, Thomas, et al. "Neural relational inference for interacting systems." International conference on machine learning. Pmlr, 2018.

[2] Linderman, Scott W., et al. "Recurrent switching linear dynamical systems." arXiv preprint arXiv:1610.08466 (2016).

[3] Dong, Zhe, et al. "Collapsed amortized variational inference for switching nonlinear dynamical systems." International Conference on Machine Learning. PMLR, 2020.

[4] Ansari, Abdul Fatir, et al. "Deep explicit duration switching models for time series." Advances in Neural Information Processing Systems 34 (2021): 29949-29961.

[5] Liu, Yongtuo, et al. "Graph switching dynamical systems." International Conference on Machine Learning. PMLR, 2023.

**Strengths And Weaknesses:**

**Strengths**
- The paper is generally very well-written, and the presented problem seems very interesting.
- The model is parameter efficient, and approximate inference is justified given the prohibitive cost of exact inference.
- The experiments show direct applications to real-world scenarios, and advantages of the proposed approach in terms of interpretability.

**Weaknesses**
- The experiments only consider very low-dimensional setups $D=2$ in all settings.
- The model might suffer from underfitting in certain complex scenarios, due to following aspects:
  - Assuming only discrete latent variables.
  - Allowing only first-order dependencies: $p(x_{t}^j | x_{t-1}^j, z_{t-1}^j)$. Could we replace the proposed distribution with a RNN?
  - Only allowing entity-level interactions, i.e. $x_{t}^j$ only sees $j$, although it can see other variables through $s_t$ via $z_t^j$, but with limitations as $s$ and $z$ are discrete. This looks very restrictive, specially in complex data such as NBA.
- **Some important baselines might be missing.** Although the aim of the method is to provide a parameter efficient model that provides data interpretability thanks to the latent space simplicity, comparisons with more powerful baselines (e.g. with continuous latent variables) should be present. See **Requested Changes**.

---

> ### Author Response · Authors · 2025-04-25
> **Response to Pthk**
>
> Thank you for your thoughtful and constructive comments. We are glad you found that the “presented problem seems very interesting”, the “model is parameter efficient”, and the “experiments show direct applications to real-world scenarios”
>
> We provide brief responses to several issues raised below. Please also see the revised PDF (key changes throughout are in red text) for detailed revisions.
>
> ## Request: Experiments with more baselines
>
> > comparisons with more powerful baselines (e.g. with continuous latent variables) should be present … to understand whether the improved interpretability comes at the cost of forecasting performance
>
> ~~Our response is *in-progress*. We are currently running some comparisons with a switching nonlinear dynamical system baseline on basketball data. We hope to have updated results very soon to share. Please look for this in the coming days.~~
>
> **Update 2025-05-05**: Please see below comment titled "Improved comparisons with SNLDS on Basketball data"
>
> We have additionally revised the related work section, to cite and discuss the paper you suggested, Graph Switching Dynamical Systems (GRASS) (Liu et al 2023).
>
> ## Issue: Experiments focus on low-dimensional (D=2) tasks
>
> > All the experiments consider 2D trajectories. Would it be possible to devise a synthetic experiment with increasing D … ?
>
> Please see new Appendix F.4 in our revised PDF manuscript, which summarizes new experiments on D=10 and D=30 feature dimensions. We have created a revised MarchingBand synthetic dataset that retains the original observations for the first two feature dimensions, and insert pure noise (Gaussian with zero mean, small variance) for other dimensions when D > 2.
>
> The results here demonstrate our current codebase can handle problems up to the highest tested value (D=30). Out of many runs, the best runs at each D still deliver high accuracy (>80%) at segmenting the true L/A/U/G/H states at the system level. Naturally, local optima problems, initialization quality, and convergence issues will increase with larger D as the problem becomes more complicated (more noise, less signal). Thus, we do see some drop in “average run” accuracy as D increases, as we might expect, especially because we keep the compute budget (number of CAVI iterations) constant across runs here.
>
> To clarify, our method has no special problem with large dimensions that is not also shared by any probabilistic model using a full covariance matrix to parameterize the likelihood of observed features. Storage scales with the number of covariance matrix parameters (D^2), and runtime scales with the cost of evaluating the PDF which requires inverting the covariance matrix, O(D^3).
>
> To scale to even larger D, we recommend considering low-rank or diagonal parameterizations of the covariance matrix. Both of these options are straightforward with our method, though the current codebase would need some edits to support them.
>
> ## Issue: The model might suffer from underfitting in certain complex scenarios
>
> We agree that the method has various limitations. We have revised our Sec. 6 to better acknowledge the points mentioned by the reviewer (discrete latents being less expressive than continuous ones, first order dependencies, limited expressivity of interactions between entities).
>
> We stress that our choice to limit cross-entity interactions to be mediated by the top-level state variable s is very intentional, so that we can have scaling that is *linear* in the number of entities rather than potentially quadratic (all pairs). For some applications, deliberately capturing cross-entity interactions may be beneficial and thus other models might be appropriate. However, for us, we think there’s a place for models like ours that can scale to 64 or 200 entities (see revised Marching Band experiments). Handling that scale would be more difficult with methods that scale quadratically in the number of entities J by modeling pairwise interactions.
>
> ## Other requested changes
>
> > The "optional exogenous covariates" mentioned in Eqs. 2.2ab might be confusing.
>
> Thanks, we have updated text below that equation to give a concrete example for the NBA basketball modeling task.
>
> > In section 2, … statement is imprecise: “If we remove the top-level system states ... our HSRMD reduces to Linderman's recurrent autoregressive HMM”
>
> Thanks, we have revised that sentence in Sec. 2 to specify that in the special case where we remove the top-level state, each entity is modeled separately by a *per-entity* rAR-HMM, as suggested. We agree this is clearer.
>
> ## Miscellaneous points
>
> > Would it be possible to try the NBA experiment replacing Eq. (2.3) with a Recurrent Neural Network instead?
>
> This is an interesting idea. We did not have time/bandwidth to try this in the response period. However, we have revised the manuscript to indicate the potential promise of this approach in Section 6.

---

> ### Author Response · Authors · 2025-04-26
> **Update on Requested Experimental Comparisons to SNLDS or RED-SDS**
>
> > Only 3 baselines are present: Agentformer and rAR-HMM in NBA; and rAR-HMM and DSARF in MarchingBand.
>
> Please note that in our original submission, we also compared to GroupNet (Xu et al. 2022) for the basketball data. That baseline is included in Table 1 and Fig. 3, along with Agentformer and rAR-HMM.
>
> ## Experimental progress on SNLDS or RED-SDS
>
> > At least 1 baseline that considers Switching Dynamical Systems should be present … consider rSLDS from Linderman et al. (2017) [2], or one of SNLDS (Dong et al. 2020) [3], RED-SDS (Ansari et al. 2021) [4]
>
> ~~Based on this suggestion, we have invested considerable time (>1 week of research effort) trying to get one of these baselines to work well. Unfortunately we must report back that we have not yet been able to produce sensible multivariate forecasts on our NBA dataset using available code as suggested above.~~
>
> ~~We used the open source RED-SDS implementation by that paper’s authors <https://github.com/abdulfatir/REDSDS>. We chose this because:~~
>
> * ~~That codebase supports both SNLDS and REDSDS as possible models.~~
> * ~~This is the newest suggested codebase in the SDS category, so it should be the most “state-of-the-art”~~
> * ~~As the newest codebase, it should be easier to install and maintain (older code is often tough to get working on newer hardware, such as the non-Intel chips of newer macbooks)~~
>
> ~~It is worth noting that in the RED-SDS paper (Ansari et al. NeurIPS 2021), there are no experiments that examine multivariate forecasting specifically; instead, there are evaluations of univariate forecasts (their Sec. 5.2) or evaluations of segmentation quality (their Sec. 5.1). That said, the repo had code clearly intended for many-step-ahead multivariate forecasting, so our use case seems supported.~~
>
> ~~We then ran experiments fitting SNLDS to our NBA data. SNLDS inherently models only entity-level time series by design, it is not originally designed to model many interacting entities. We first tried the Concat entity-to-system strategies (as described in Sec. 5.1 of our paper) to adapt it to model many entities.~~
>
> ~~We then verified training worked well by~~:
> * ~~examining trace plots to verify the ELBO training objective converged after many epochs (see Fig. E.1 in our revised PDF)~~
> * ~~verifying that reconstructions of past data looked reasonable~~
> * ~~verifying that hyperparameter settings (e.g. model size, optimization settings) were sensible given the documented experimental configurations for past experiments in the repo~~
>
> ~~After what appeared to be successful training, we produced multivariate forecasts on the test set. Forecasting ran without issue for the Concat. strategy models, yet produced what appeared to be quite poor forecasts. Visualizing a single player’s forecast, we see the SNLDS forecast seems to often go in an unrelated direction with quite high velocity. We’ve included plots in a new Appendix (see Fig E.2 in our revised PDF)~~
>
> ~~We quickly tried RED-SDS as well, and found a similar story (training seems to have gone well, but multivariate forecasting delivers quite poor results).~~
>
> ~~We further tried SNLDS with the Independent entity-to-system strategy. In this case, running the package’s provided forecasting code gave us a clear error: a dimension mismatch in a matrix multiply. In particular, the code seems to assume that the dimensionality of the prediction interval matches the dimensionality of the latent continuous state, which does not make much sense.~~
>
> ~~We have spent considerable time just getting to the point of identifying this bug; we have not managed to fix it yet.   We hope reviewers can appreciate the difficulty of debugging research code written by a different person.~~
>
> Update 25-05-06: we have fixed a bug and now have sensible forecasting results to share for the requested SNLDS baseline on the basketball dataset. See the freshly revised PDF, where purple text denotes new content in Sec. 5.2 added now in May 2025. To summarize the new content in Fig 3, our analysis of basketball data suggests that SNLDS applied individually to each player yields quantitative forecasts that are worse than our HSRDM.

---

> ### Author Response · Authors · 2025-04-26
> **Outlook: Where to go from here on handling the additional experiments request?**
>
> ~~Based on the comment above, where our efforts to compare to SNLDS and RED-SDS were unsuccessful, we ask reviewers how we should proceed. Perhaps with some more time (another week or two), we could fix the bug identified above and acquire reasonable forecasts from SNLDS or RED-SDS. Or we could pursue another different baseline (again with more time needed).~~
>
> ~~However, stepping back, we would suggest that such extra intense effort perhaps isn’t needed. These SNLDS and RED-SDS baselines are not intended for our *team dynamics* modeling applications. We are happy to acknowledge more forcefully in the paper that there may be cases for *individual entity* modeling where these non-linear SLDS papers have clear advantages over our style of model in terms of capacity. However, for modeling a team of coordinated entities, multi-entity models have clear advantages over single-entity models, as demonstrated both in the ablation experiments of this paper and in the larger literature on multi-agent models.~~
>
> ~~To revisit again the reviewer’s original requests:~~
>
> ~~> comparisons with more powerful baselines (e.g. with continuous latent variables) should be present~~
>
> ~~Please note that DSARF does have continuous latent variables.~~
>
> ~~> It would be very important to include additional baselines to understand whether the improved interpretability comes at the cost of forecasting performance.~~
>
> ~~We suggest that existing results in our submission already prove the reviewer’s point. The GroupNet baseline *for basketball* does already deliver better forecasts than our approach. However, as we argue in the paper, GroupNet cannot achieve our interpretability goals, since it cannot offer discrete segmentations.~~
>
> ~~We would be curious to hear your updated thoughts. Again, in the limited two week response period we are sorry that we couldn’t achieve all the experiments you were hoping for. We think we’ve made a good faith effort and we hope you consider the considerable time costs of additional experiments in your decision making.~~
>
> Update 25-05-06: we have fixed a bug and now have sensible forecasting results to share for the requested SNLDS baseline on the basketball dataset. See the freshly revised PDF, where purple text denotes new content in Sec. 5.2 added now in May 2025. To summarize the new content in Fig 3, our analysis of basketball data suggests that SNLDS applied individually to each player yields quantitative forecasts that are worse than our HSRDM.

---

> ### Author Response · Authors · 2025-05-06
> **Update for reviewer PthK in May 2025: Improved comparisons with SNLDS on Basketball data**
>
> **Update 25-05-06**: After substantial debugging, we now have sensible forecasting results to share for the requested SNLDS baseline on the basketball dataset. See the freshly revised PDF, where purple text denotes new content in Sec. 5.2 added now in May 2025. To summarize the new content in Fig. 3 and Table 1(a), our analysis of basketball data suggests that SNLDS applied individually to each player yields quantitative forecasts that are worse than our HSRDM.
>
> To revisit the reviewer’s overall requests:
>
> > comparisons with more powerful baselines (e.g. with continuous latent variables) should be present
>
> As requested, we now include SNLDS results in the latest revision of Sec. 5.2 (see purple text for changes since the last revision in late April).
>
> Please note that DSARF (included in original submission as a baseline in several other experiments) also does have continuous latent variables.
>
> > It would be very important to include additional baselines to understand whether the improved interpretability comes at the cost of forecasting performance.
>
> In our latest revised experiments in Sec. 5.2, we find that independent SNLDS models for each player, despite flexible non-linear transition dynamics, do not deliver better forecasts than our HSRDM. This suggests that the HSRDM’s ability to model group-level dynamics as well as entity-level dynamics is important, while flexible non-linear models like SNLDS applied to model each entity individually, without any group-level coordination, are not always enough for high-quality forecasts.
>
> Stepping back, we also suggest that other existing results in our submission already prove the reviewer’s larger point about tradeoffs between interpretability and forecasting performance. The GroupNet baseline for basketball does already deliver better forecasts than our approach. However, as we argue in the paper, GroupNet cannot achieve our interpretability goals, since it cannot offer discrete segmentations.
>
> In closing, we thank the reviewer for their useful suggestions, which have improved our manuscript. We hope you'll reach out if there's anything further we can clarify.

---

> ### Comment · Reviewer_Pthk · 2025-05-09
> **Response to authors**
>
> Dear Authors,
>
> Thank you very much for your detailed response to my review, and apologies for misunderstandings in terms of missing baselines. After reading your rebuttal and revisiting the revised manuscript, I am satisfied with the current version of the paper.
>
> In particular, I appreciate the authors' efforts in including experiments with baselines based on Switching Dynamical Systems (SDS). I believe it was important to compare your method with models that incorporate continuous latent variables, as these can be expected to offer more expressive latent representations. The additional experiments clearly demonstrate that, despite its simplicity, the proposed model is effective both in terms of interpretability and forecasting performance.

---

### Review · Reviewer_p3rW · 2025-03-13

**Summary Of Contributions:**

This paper introduces a novel modeling framework, called Hierarchical Switching Recurrent Dynamical Models (HSRDM), designed to infer coordinated group dynamics from multiple interacting time series. Key contributions include:
1. This paper proposes a hierarchical latent switching-state structure, which includes a system-level Markov chain that represents global state dynamics influencing individual entities, and entity-level chains governing individual trajectories.
2. This work also proposes system-level states influence entity dynamics (top-down), while observations from entities influence system-level transitions (bottom-up), improving interpretability and forecasting accuracy.
3. Experiments showing the model’s effectiveness on synthetic datasets (FigureEight, MarchingBand) and real-world scenarios (Basketball player trajectories, soldier behavior), demonstrating competitive forecasting performance and interpretable insights.

**Audience:**

Yes

**Broader Impact Concerns:**

The paper currently does not explicitly address potential ethical concerns or misuse scenarios. A broader impact statement would strengthen the submission, explicitly considering issues such as privacy concerns when modeling individual-level data or unintended biases when deploying group dynamic models for sensitive tasks (e.g., military or surveillance applications).

**Claims And Evidence:**

Yes

**Requested Changes:**

The following changes are recommended, primarily to strengthen the work:
1. Please include experiments with higher-dimensional observational data or a significantly larger number of entities to demonstrate scalability and robustness. (Strengthens work)
2. Clearly highlight conditions under which the proposed HSRDM would significantly outperform or underperform relative to neural network baselines such as transformers or RNN-based models.
(Strengthens work)
3. Please add a detailed ablation study explicitly evaluating the individual impacts of top-down vs. bottom-up recurrence across all experiments. (Strengthens work)
4. The description of NRI (Kipf et al., 2018) in the related work is not accurate. It always works with the assumption of a fully connected structure. The connectivity of the graph does not have any influence on the runtime. Actually, it works better with dense graphs.
4. As some idea look similar to the work in the field of relational inference, it would be better to include them in the related work. E.g.: Node-level variational autoencoder for efficiency: A. Wang and J. Pang. Structural inference with dynamics encoding and partial correlation coefficients. ICLR 2024.

**Strengths And Weaknesses:**

Strengths:
1. Clear and novel integration of top-down and bottom-up feedback mechanisms, effectively capturing complex group dynamics.
2. Efficient variational inference algorithm with runtime linear in entity count, enabling scalability.
3. Strong empirical validation with both synthetic and real datasets.
4. Practical interpretability of learned states and dynamics, suitable for applications requiring domain knowledge integration .

Weaknesses:
1. Potential scalability concerns for datasets with higher dimensionality of observations.
2. Experiments largely focus on moderately sized problems (in terms of entities and complexity); larger-scale scenarios are less explored.

---

> ### Author Response · Authors · 2025-04-25
> **Response to reviewer p3rW**
>
> Thank you for your thoughtful and constructive comments.
>
> ## Request 1a: experiments with higher-dimensional observational data
>
> Please see new App. F.4 in our revised PDF manuscript, which summarizes new experiments on D=10 and D=30 feature dimensions. We created a revised MarchingBand synthetic dataset that retains the original observations for the first two feature dimensions, and inserts pure noise (Gaussian with zero mean, small variance) for other dimensions when D > 2.
>
> Results demonstrate our codebase can handle problems up to the highest tested value (D=30). Out of many runs, the best runs at each D still deliver high accuracy (>80%) at segmenting the true L/A/U/G/H states at the system level. Naturally, local optima problems, initialization quality, and convergence issues will increase with larger D as the problem becomes more complicated (more noise, less signal). Thus, we do see some drop in “average run” accuracy as D increases, as we might expect, especially because we keep the compute budget (number of CAVI iterations) constant across runs here.
>
> To clarify, our method has no special problem with large dimensions that is not also shared by any probabilistic model using a full covariance matrix to parameterize the likelihood of observed features. Storage scales with the number of covariance matrix parameters (D^2), and runtime scales with the cost of evaluating the PDF which requires inverting the covariance matrix, O(D^3).
>
> ## Request 1b: include experiments with a significantly larger number of entities
>
> Please see new Appendix F.3 in our revised PDF manuscript. This appendix summarizes a new experiment on the MarchingBand synthetic dataset, where we used a similar generative process but created 200 entities rather than 64 as in the original experiment. The median accuracy at recovering system states was 100% across 5 trials (5 separate random initializations run to convergence). This demonstrates our method and codebase can handle a few hundred entities.
>
> ## Request 2: highlight conditions under which the proposed HSRDM would significantly outperform or underperform relative to neural network baselines such as transformers or RNN-based models
>
> We have added a paragraph named “When to favor the HSRDM over alternatives?” to Sec. 6 (Discussion & Conclusion) of the revised PDF.
>
> We suggest our approach is most beneficial when the entities being modeled have some clear top-down coordination known to a domain expert, and that both explanatory and prediction purposes of the model are relevant. See the added paragraph for more details.
>
> ## Request 3: ablation study evaluating individual impacts of top-down vs. bottom-up recurrence
>
> We agree that such ablations are valuable: both special cases are discussed at the end of Sec. 2. When the top-down coordination of the system state is removed, we recover a separate rAR-HMM for each entity (labeled as either “No system state (rAR-HMM)” or “rAR-HMM independent” in several experiments). Removing the bottom-up recurrence is another natural baseline; it does not exactly match any previous published model, so we refer to it as “No recurrent feedback”).
>
> We hope our latest revision fully addresses this issue across 3 datasets. The specific places to find each result are enumerated below. In fact, we already included most of the requested experiments in our original submission, though perhaps they could have been better labeled. As such, we slightly adjusted the labeling in the revised PDF to be more consistent across experiments:
>
> FigureEight ablations:
>
> * Remove top-down: See existing results for “rAR-HMM Independent” in Fig 2
> * Remove bottom-up: See *new* results for “No recurrent feedback” in Fig 2 and in App Fig D.1.
>
> Basketball ablations
>
> * Remove top-down: See existing results for “No system state (rAR-HMM) in Fig 3 and Tab. 1
> * Remove bottom-up: See existing results for “No recurrent feedback” in Fig 3 and Tab 1
>
> Marching band ablations
>
> * Remove top-down: See existing results for “No system state (rAR-HMM)” in Fig 4 and Tab 3
> * Remove bottom-up: See existing results for “No recurrent feedback” in Fig 4 and Tab 3
>
> ## Request 4: revise description of NRI (Kipf et al. 2018)
>
> Thanks for this helpful request. We have revised that paragraph to make clear that the work of Kipf et al. focuses on learning “interaction types” for each edge in a fully-connected graph, not the existence of the edges.
>
> We apologize for the faulty description. Our error came from misunderstanding of that paper’s Figure 1 and Figure 7 and various tables in that work that used the method names like “NRI (full graph)” and “NRI (learned graph)”. Hopefully the issue is now resolved.

---

> ### Author Response · Authors · 2025-04-25
> **Response to reviewer p3rW (part 2 of 2)**
>
> ## Request 5: include work in the field of relational inference
>
> Thanks, we have cited and discussed Wang and Pang (ICLR 2024) work in the revised Related Work section. It is an interesting approach for estimating the interaction graph that demonstrates some practical scalability gains over older NRI methods like Kipf et al. It differs from our approach, as it seems its primary purpose is graph estimation rather than forecasting or interpretable segmentation.
>
> ## Broader impacts
>
> Thanks for this suggestion. We have added a Broader Impacts section to the revised manuscript PDF, after Sec 6 yet before References.

---

### Review · Reviewer_iSuN · 2025-04-13

**Summary Of Contributions:**

In this paper, the authors introduce an extension to recurrent auto-regressive hidden Markov Models  that allows for segmentations of systems involving multiple interacting agents. Specifically, the authors introduce a hierarchical latent state that models the group dynamics of the agent along with agent level latent states. This hierarchy, combined with a recurrent connection from the observations, allows for a simple, yet expressive generative model.

**Audience:**

Yes

**Claims And Evidence:**

Yes

**Requested Changes:**

I have no requested changes!

**Strengths And Weaknesses:**

# Strengths

I think this an absolutely fantastic paper and was written beautifully! The motivation for the approach was expertly told and the method was described concisely. I also found the breadth of experiments to be amazing and highlights the potential of this model.

# Weaknesses

There are no major weaknesses to this work in my opinion. I think the only thing that I would add is from the writing, it is not clear whether the recurrence parameters, $\phi, \psi$ are optimized in the varational M-step.

---

> ### Author Response · Authors · 2025-04-25
> **Response to reviewer iSuN**
>
> Thank you very much for your positive comments. We are grateful that you appreciate the motivation and the experimental results.
>
> > not clear whether the recurrence parameters …  are optimized in the varational M-step.
>
> We have revised the M-step description in Sec. 3 (see revised PDF with changes in red) to clarify that the parameters of the recurrent feedback functions can be updated in the M-step when they are learnable.
>
> In all our experiments, we were able to use simple, parameter-free recurrence functions. For example, the recurrence needed in our basketball analysis uses player locations and out-of-bounds indicators. So in our experiments, there is no need to do specific M-step updates of these parameters. Only the transition and emission parameters are updated in the M-step.

---

### Author Response · Authors · 2025-05-06
**Update in May 2025: Improved comparisons with SNLDS on Basketball data**

Update 25-05-06: we have fixed a bug and now have sensible forecasting results to share for the requested SNLDS baseline on the basketball dataset. See the freshly revised PDF, where purple text denotes new content in Sec. 5.2 added now in May 2025. To summarize the new content in Fig 3, our analysis of basketball data suggests that SNLDS applied individually to each player yields quantitative forecasts that are worse than our HSRDM.

---

### Comment · Editors_In_Chief · 2025-07-02

The Editors in Chief reviewed this paper and its IRB approval, and do not judge that it violates TMLR's principles on ethical publishing.

---

### Decision · Action_Editor_KH1r · 2025-07-16

**Recommendation:** Accept with minor revision

**Additional Comments:**

Please include the updated experimental results as well as promised revisions in the final camera ready.

Since this paper also includes experiments on soldier tracking data (which is very sensitive), I recommend editing the text and mentioning about the ethics and IRB approval explicitly in broader impact section, e.g., if allowed, provide disclosure regarding the approval body of the IRB. Please also consult with TMLR EiCs should further clarifications regarding ethics declaration is required for the camera ready.

**Audience:**

Yes

**Audience Explanation:**

Researchers on probabilistic inference and sequence modelling would be interested in reading this paper.

**Claims And Evidence:**

Yes

**Claims Explanation:**

This paper proposes an extension to auto-regressive HMMs mainly for time-series segmentation tasks. The idea is to use a hierarchical latent variable structure (discrete, like "switches") which captures both system-level and entity-level dynamic switching behaviour. Experiments show the model’s good performance on synthetic datasets and real-world scenarios (Basketball player trajectories, soldier behaviour).

Reviewers overall welcomed the contribution and recognised the clarity of the presentation. They also thought that the experiments are comprehensive, although they also raised concerns regarding the small scales (e.g., dimensionality and dataset size) for many of the example applications in the experiments. Additional experiments were conducted during the revision stage which largely addressed the reviewers' concerns.